

# *Bosminopsis deitersi* (Crustacea: Cladocera) as an ancient species group: a revision

Petr G. Garibian[1,*], Dmitry P. Karabanov[2], Anna N. Neretina[1], Derek J. Taylor[3] and Alexey A. Kotov[1,*]

[1] Laboratory of Aquatic Ecology and Invasions, A.N. Severtsov Institute of Ecology and Evolution of Russian Academy of Sciences, Moscow, Russia
[2] Laboratory of Fish Ecology, I.D. Papanin Institute for Biology of Inland Waters of Russian Academy of Sciences, Borok, Yaroslavl Area, Russia
[3] Department of Biological Sciences, The State University of New York at Buffalo, Buffalo, New York, United States
* These authors contributed equally to this work.

## ABSTRACT

Water fleas (Crustacea: Cladocera) of the Family Bosminidae have been studied since the founding of paleolimnology and freshwater ecology. However, one species, *Bosminopsis deitersi*, stands out for its exceptional multicontinental range and broad ecological requirements. Here we use an integrated morphological and multilocus genetic approach to address the species problem in *B. deitersi*. We analyzed 32 populations of *B. deitersi* s. lat. Two nuclear and two mitochondrial loci were used to carry out the bGMYC, mPTP and STACEY algorithms for species delimitation. Detailed morphological study was also carried out across continents. The evidence indicated a widely distributed cryptic species in the Old World (*Bosminopsis zernowi*) that is genetically divergent from *B. deitersi* s.str. We revised the taxonomy and redescribed the species in this complex. Our sampling indicated that *B. zernowi* had weak genetic differentiation across its range. A molecular clock and biogeographic analysis with fossil calibrations suggested a Mesozoic origin for the *Bosminopsis deitersi* group. Our evidence rejects the single species hypothesis for *B. deitersi* and is consistent with an ancient species group (potentially Mesozoic) that shows marked morphological conservation. The family Bosminidae, then, has examples of both rapid morphological evolution (Holocene *Bosmina*), and morphological stasis (*Bosminopsis*).

## INTRODUCTION

*Frey (1962)* demonstrated morphological stasis for the water fleas (Cladocera) based on paleolimnological records from the Quaternary. He later based the paradigm of "non-cosmopolitanism" (*Frey, 1982*, *1987b*) on this apparent long-term stability in morphology. According to non-cosmopolitanism, geographic differentiation occurred mainly due to vicariant events related to the disruption of Pangaea and the dispersal barriers imparted by subsequent continental drift. The process (often termed "continental endemism") now has strong support among "traditional" taxonomists (*Van Damme & Kotov, 2016*;

Corresponding author
Alexey A. Kotov,
alexey-a-kotov@yandex.ru

*Smirnov & Kotov, 2018*; *Neretina, Kotov & Van Damme, 2019*) and molecular ecologists (*Xu et al., 2009*; *Heads, 2012*).

Frey's early insights on non-cosmopolitanism made the Cladocera (*Frey, 1982*, *1987a*) a model group for freshwater animals. However, a transcontinental distribution for many freshwater taxa persists. One of these species reported from many continents is *Bosminopsis deitersi Richard, 1895* (Cladocera: Bosminidae). After the first description of *B. deitersi* from Rio de La Plata (*Richard, 1895*), the species was found in many tropical (*Daday, 1903*; *Brehm, 1913*, *1939*; *Rahm, 1956*; *Dumont, 1981*; *Idris, 1983*; *Collado, Fernando & Sephton, 1984*; *Dumont, 1986*; *Tanaka & Ohtaka, 2010*; *Korovchinsky, 2013*; *Kotov et al., 2013*) and temperate (*Linko, 1901*; *Birge, 1918*; *Ueno, 1932*; *Pirozhnikov, 1937*; *Ueno, 1937a*; *Ueno, 1937b*, *1940*; *Tanaka, 2000*; *Jeong, Kotov & Lee, 2014*; *Beaver et al., 2018*) regions. Citing minor morphological differences from *B. deitersi* (see checklist below), several authors described regional taxa. *Bosminopsis zernowi Linko, 1901*, found in European Russia, was the second taxon to be named in this group. *Burckhardt (1909)* concluded that there are three alternative views of the group's diversity: 1) at least eight local forms ("Lokalväriataten"), 2) multiple independent species, and 3) a single broadly distributed taxon, *B. deitersi*. Later he advocated 11 extant "taxa" in the group (*Burckhardt, 1924*).

Burckhardt's view failed to gain support, with most authors recognizing that *Bosminopsis* is a monotypic genus (*Krasnodebski, 1937*; *Ueno, 1937a*; *Ueno, 1937b*; *Chiang & Du, 1979*; *Yoon, 2010*). *Behning (1941)* and Manujlova (*Manujlova*) regarded some forms described by Burckhardt as regional subspecies but gave very obscure diagnoses. At the end of the 20th century, researchers of tropical populations (*Frey, 1982*; *Smirnov & Timms, 1983*; *Michael & Sharma, 1988*; *Smirnov, 1995*; *Sanoamuang, 1997*) also assigned specimens to a single taxon, *B. deitersi*. Indeed, Korinek (1984) directly stated that "*Bosminopsis deitersi* was regarded as the only species within the genus". However, in the last third of the 20th century, two species of *Bosminopsis* beyond the *B. deitersi* group were found in the Amazon River basin (*Brandorff, 1976*; *Rey & Vasquez, 1989*).

*B. deitersi*, as presently described, has an unusually broad biogeographic range and ecological preference for a cladoceran species. *Wolsky (1932)* wrote that *Bosminopsis deitersi* prefers "warm water". However, *Pirozhnikov (1937)* detected *Bosminopsis* in the high latitude waters of the Ob and Yenisei river deltas. *Kotov (1997b)* suggested that these North Eurasian populations belong to a separate taxon from *B. deitersi*, *B. zernowi* Linko, 1900.

An integrated approach, which combines molecular phylogeny, phylogeography, formal biogeography, and morphological analysis has advanced the taxonomy of several difficult groups (*Dayrat, 2005*; *Padial & de La Riva, 2010*). Here we use an integrated approach to address the taxonomy of the *B. deitersi* group. We find evidence that the group contains several related species with modest geographic ranges and weak morphological differentiation. We reconstruct this group's evolutionary history and provide evidence for the antiquity and morphological conservation of the genus *Bosminopsis*. We also redescribe *B. deitersi* and *B. zernowi* and analyze their synonyms.
## MATERIALS & METHODS

### Ethics statement

Field collection in public property in Russia does not require permission. Samples from South Korea were collected in the frame of cooperation between A.A. Kotov and the National Institute of Biological Resources of Korea and does not require special permission. The sample from Arkansas, USA was obtained from collections resulting from a previous NSF grant. The samples from Japan, China, and Thailand were provided by our colleagues having permissions to collect them due to their scientific activity in the governmental institutes of the corresponding countries. Formalin samples from Brazil were kept in the Collection of Zoological Museum of Moscow State University for a long time, they were collected before Brazil introduced very strict regulations for sampling. The species were not assessed as endangered at the time of collection and are currently not subject to specific regulations, however all efforts were taken to ensure that the collection and preservation of animals was performed with due consideration of their welfare. The number of individuals taken did not represent a significant proportion of the population present at each site.

### Sample collection and morphological analyses

Samples from different localities were collected via small-sized plankton nets (with mesh size 50 μm) and fixed via 4% formaldehyde or 96% ethanol in the fields, using conventional techniques. All samples were initially examined using a stereoscopic microscope LOMO. Individuals of *Bosminopsis* were initially identified via available references using morphological features (*Kotov, 1997a*, *1997b*; *Rogers et al., 2019*). Existing museum samples were used for morphological analysis (see the list of material in Table S1).

The morphology of populations from the Neotropics and the Palaearctic was examined in detail to assess the presence of taxonomically significant characters. Specimens of *Bosminopsis* from presorted formalin and alcohol samples were selected under a binocular stereoscopic microscope LOMO. They were then studied in toto under optical microscopes (Olympus BX41 or Olympus CX 41) in a drop of glycerol formaldehyde or a glycerol-ethanol mixture. Then, at least two parthenogenetic females, two ephippial females, and two males (if available) from each locality were dissected under a stereoscopic microscope for appendages and postabdomens. Drawings were prepared using a camera lucida attached to the optical microscopes.

Some individuals from Neotropical and Palaearctic localities were dehydrated in an ethanol series (30%, 50%, 70%, 95%) and 100% acetone and then dried from hexametyldisilazane. Dried specimens were mounted on aluminum stubs, coated by gold in S150A Sputter Coater (Edwards, West Sussex, United Kingdom), and examined under a scanning electron microscope (Vega 3 Tescan Scanning Electron Microscope; TESCAN, Czech Republic).

### DNA extraction, amplification and sequencing

Only alcohol samples were used for the genetic analysis. Each specimen was identified by morphological characters (Table S1). Genomic DNA was extracted from single adult

females using the Wizard Genomic DNA Purification Kit (Promega Corp., Madison, WI, USA) and QuickExtract DNA Extraction Solution (Epicentre by part Illumina, Inc., Madison, WI, USA) using manufacturer's protocols. Two mitochondrial and two nuclear markers were investigated here: (1) the 5′-fragment of the first subunit of mitochondrial cytochrome oxidase (COI)–a protein-coding marker widely used in DNA barcoding (Hebert et al., 2003); (2) the 5′-fragment of the mitochondrial 16S rRNA gene (16S) with a mosaic of highly conservative duplexes and variable loops (Yang et al., 2014); and (3–4) 5′-fragments of the nuclear ribosomal genes (18S rRNA and 28S rRNA). Each fragment contains both long conservative portions and two variable domains. Although these nuclear markers are predominantly used for divergent taxa (Hovmoller, Pape & Kallersjo, 2002), they are informative at the species level for many microcrustaceans (Karabanov et al., 2018).

Primers used for amplification are listed in Table 4. Polymerase chain reactions (PCR) were carried out in a total volume of 20 μl, consisting of 2 μl of genomic DNA solution, 1 μl of each primer (10 mM), 6 μl of double-distilled $H_2O$ and 10 μl of ready-to-use PCR Master Mix 2X solution (Promega Corp., Madison, WI, USA). PCR products were visualized in a 1.5% agarose gel stained with ethidium bromide and purified by QIAquick Spin Columns (Qiagen Inc., Valencia, CA, USA). The PCR program included a pre-heating of 3 min at 94 °C; 40 cycles (initial denaturation of 30 s at 94 °C, annealing of 40 s at a specific temperature, an extension of 80 s at 72 °C); and a final extension of 5 min at 72 °C (Table 4). Each PCR product was sequenced bi-directionally on an ABI 3730 DNA Analyzer (Applied Biosystems, Foster City, CA, USA) using the ABI PRISM BigDye Terminator v.3.1 kit at the Syntol Co, Moscow. The authenticity of the sequences was verified by BLAST comparisons with published cladoceran sequences in mBLAST (Boratyn et al., 2013).

The sequences from this study were submitted to the NCBI GenBank database for 16S acc. no. MT757174–MT757231, for COI acc. no. MT757459–MT757473, for 18S acc. no. MT757232–MT757274 and for 28S acc. no. MT757314–MT757388.

## Population analysis, alignment and phylogenetic analysis

Alignment of sequences from each locus was carried out using the MAFFT v.7 algorithm (Katoh, Rozewicki & Yamada, 2019) available on the server of the Computational Biology Research Center, Japan (http://mafft.cbrc.jp). For the protein-coding gene COI, we used translation alignment with the FFT-NS-i strategy. For alignment of the ribosomal-coding loci, we used the Q-INS-i strategy (secondary structure is considered by this algorithm). Linking sequences and their partitioning for subsequent analyses were made in SequenceMatrix v.1.8 (Vaidya, Lohman & Meier, 2011).

Nucleotide diversity analysis (Nei & Kumar, 2000) and neutrality tests were carried out using DnaSP v.6.12 (Rozas et al., 2017). We applied the Fs (Fu, 1997) and D (Tajima, 1989) tests to confirm neutrality and describe demographic processes in Bosminopsis population (Ramirez-Soriano et al., 2008; Garrigan, Lewontin & Wakeley, 2010). To determine the most probable demographic model for Bosminopsis sequences, we performed a coalescent simulation for each locus (1,000 replications) in DnaSP v.6.12 (Rozas et al., 2017) using five

demographic models (Standard Neutral Model, Population Growth, Population Decline, Population Bottleneck, Population Split/Admixture). The best model was selected based on the Theta-W (theta with Watterson) estimator (*Watterson, 1975*; *Nei & Kumar, 2000*).

The best-fitting models of the nucleotide substitutions for each locus and for linked data were selected using ModelFinder v.1.6 (*Kalyaanamoorthy et al., 2017*) at the Center for Integrative Bioinformatics Vienna web-portal, Austria (http://www.iqtree.org) (*Trifinopoulos et al., 2016*) based on minimal values of the Bayesian information criteria (BIC) (*Schwarz, 1978*). The BIC model parameters were almost identical to those obtained using the corrected Akaike's information criterion, AICc.

For the *COI* locus, the substitution model was partitioned by the nucleotide position of codons (1st, 2nd, 3rd). We used the multi-taxon coalescence model "star" in BEAST2 (*Heled & Drummond, 2010*) with partitioned models (*Chernomor, Haeseler & Minh, 2016*). Phylogenetic reconstructions based on the maximum likelihood (ML) and Bayesian (BI) methods were made for each gene separately, the full set of mitochondrial genes, the full set of nuclear genes, and for all "unlinked" genetic data. We included sequences with incomplete or missing data as exclusion can reduce the accuracy of phylogenetic reconstruction (*Molloy & Warnow, 2018*).

We used the IQ-TREE v.1.6.9 algorithm (*Nguyen et al., 2015*) via a web-portal CIBIV, Austria for ML tree estimation. Each set of sequences was analysed based on the best model found automatically by the W-IQ-TREE (*Trifinopoulos et al., 2016*). To estimate the branch support values, we used UFboot2 (*Hoang et al., 2018*). The Topology of the ML trees was evaluated based on PhyML SH-like tests (*Shimodaira, 2002*), performed in the block Building Phylogenetic Tree in uGENE v.34 (*Okonechnikov, Golosova & Fursov, 2012*). BI analysis was performed in BEAST2 v.2.6.2 (*Bouckaert et al., 2019*), with all of the parameters of the substitution model using the BIC criterion from BEAUti v.2.5.2 (*Drummond et al., 2012*). In each analysis, we conducted four independent runs of MCMC (40M generations and a sampling interval of 10k generations), with effectiveness control in Tracer v.1.7 (*Rambaut et al., 2018*). A consensus tree based on the maximum clade credibility (MCC) was obtained in TreeAnnotator v.2.5.2 (*Drummond et al., 2012*) with a burn-in of at least 20%. Because the main clades for BI and ML were congruent, we presented the BI trees, with ML branch support/ BI posterior probabilities for key nodes.

A haplotype network was constructed for *Bosminopsis zernowi* (the most sampled taxon in this study) in popART v.1.7 with the Integer Neighbor-Joining Network algorithm (*Leigh, Bryant & Nakagawa, 2015*) and minimal reticulation tolerance.

## Cybertaxonomic species delimitation based on DNA data

Our approach to cybertaxonomic taxon delimitation was described in a previous paper (*Kotov et al., 2021*). An integrated approach based on genetic species delimitation combined with "traditional" morphological taxonomy was used to estimate the species richness. We used the bGMYC, mPTP and STACEY algorithms for species delimitation (*Carstens et al., 2013*).

The general mixed Yule-coalescent model (GMYC) was made to assign analyzed individuals to the species according to ultrametric time trees derived from single-locus data (*Pons et al., 2006*). But the "classical" GMYC has limitations (*Lohse, 2009*). We used the Bayesian GMYC model in the 'bGMYC' package (*Reid & Carstens, 2012*) for the statistical language "Microsoft R-Open and MKL" 64-bit v.3.5.3 (http://mran.microsoft.com). Ultrametric trees for each mitochondrial and nuclear datasets were evaluated in BEAST2. For MCMC, we used 50M generations with a sampling interval of 50k trees. We used Tracer v.1.7 to evaluate the convergence of parameters (based on ESS>200). Sequences of *Triops* and *Bosmina* were used as outgroups. Sorting, re-rooting of the trees and outgroup deletion was carried out in "R" according to the script of *Sweet et al. (2018)*. For the bGMYC analysis, we randomly selected 100 ultrametric trees from the 1,000 trees after burn-in from BEAST2. The results were accepted as statistically significant at a modified $P > 0.99$ level.

Analysis of Multi-rate Poisson Tree Processes (mPTP) (*Kapli et al., 2017*) was performed on the web-server of Heidelberg Institute for Theoretical Studies (http://mptp.h-its.org/). As the input trees, we used the phylogenetic BI trees from BEAST2 and the ML-tree obtained using W-IQ-TREE for each locus. Delimitation results were congruent for separate loci and were composed of mitochondrial and nuclear datasets.

The combined species tree estimation and species delimitation analysis for STACEY (Species Tree And Classification Estimation, Yarely) (*Jones, 2017*), was made in BEAST2. Genealogical relationships were estimated by STACEY with four independent generations (50M generations of MCMC, sampling of every 10k generation) after incorporating the suggestions from an initial run. STACEY log files were examined with Tracer v.1.7 to evaluate the convergence of parameters based on ESS > 400. Supports for the tree topologies estimated by STACEY were examined by constructing a maximum clade credibility tree using TreeAnnotator v2.6 (part of the BEAST2) after discarding half of all estimated trees. Species delimitations based on the trees estimated by STACEY were assessed using the Jones' java-application speciesDA: http://www.indriid.com/software.html.

## Phylogenetic reconstruction and molecular clock

Two approaches were used for molecular clock estimation. A strict molecular clock (*Drummond & Bouckaert, 2015*). was based on the assumption of a relatively regular mutation rate in mitochondrial genes. The speed of mutation accumulation differs among organisms. For the crustaceans the rate is ca. 0.11–2.4% per MYA (*Knowlton & Weigt, 1998*; *Schubart, Diesel & Hedges, 1998*; *Schwentner et al., 2013*; *Bekker et al., 2018*). Apparently this is a very rough estimation (*Schwartz & Maresca, 2006*). An alternative approach uses paleontological data to calibrate "molecular clocks".

To estimate the probability of molecular clock-like data, we applied a Maximum Likelihood method implemented in MEGA-X v.10.1.8 (*Kumar et al., 2018*). A Maximum Likelihood substitution model was estimated for each locus (separately for each nucleotide position for translated genes, and jointly for non-translated fragments). We used the best

model as selected by the lowest BIC (Bayesian Information Criterion) scores and an ML tree with *Bosmina* as the outgroup.

For determination of the relative rate of substitutions, we used both paleontological information (*Kotov & Taylor, 2011*) and points based on molecular phylogenetic data (*Schwentner et al., 2013*). As calibration points (with 15% standard deviations), we used the following estimates: *Triops*/ all groups 340 MYA, *Daphnia*/ *Simocephalus* 145 MYA, *Cyclestheria* groups 120-70 MYA, and *Bosmina* / *Bosminopsis* without an exact date. The age of lineage differentiation according to a strict molecular clock model was estimated in BEAST v.1.10.4 (*Suchard et al., 2018*) with a Yule speciation model as the most proper for datasets with several potential species (*Gernhard, 2008*). Four independent runs of 50M generations were done, with each 100k tree sampled. Subsequent analysis was performed as above for BI following the recommendations (*Barido-Sottani et al., 2018*).

### Phylogeographic reconstructions

To test phylogeographic models, we used the packet BioGeoBEARS (*Matzke, 2013*) with the integrated statistical package of the "R" language in RASP4 v.4.2 (*Yu et al., 2015*). The data set was composed based on the maximum number of geographic localities and representation of all phylogenetic lineages revealed by cybertaxonomic methods. We estimated a mitochondrial phylogenetic tree based on sequences of two genes (*COI* and *16S*) for *Bosminopsis* cf. *deitersi*. Objective software limitations allowed us to analyze only 27 sequences from five phylogenetic lineages and seven main geographic regions.

We tested six biogeographic models in BioGeoBEARS (standard dispersion-vicariant, and those with a correction for speciation events +J), estimated according to the AICc_wt criterion (*Matzke, 2014*). A phylogeny for RASP4 was reconstructed in BEAST2. In each analysis, we conducted four independent runs of MCMC (40M generations, with a sampling interval of 10k generation). The best model according to the maximum AICc_wt value was DIVALIKE+J, which takes into consideration new lineage origin upon colonization by a founder without the existence of a widely distributed ancestor (*Clark et al., 2008*). For estimates of the age of historical processes, we used an outgroup "calibration". Palaeoreconstruction was performed in GPlates v.2.2 (*Muller et al., 2018*) with PALEOMAP PaleoAtlas v.3 by Christopher R. Scotese (https://www.earthbyte.org/paleomap-paleoatlas-for-gplates).

## ABBREVIATIONS

*Abbreviations for collections*
MGU ML – collection of Zoological Museum of Moscow State University.

*Abbreviations in illustrations and text*
I–V – thoracic limbs I–V;
dag – distal armature of gnathobase;
dis – distal setae of exopodite;
ejh – ejector hooks on limb I;
epp – epipodite;

ext – exopodite;
fpl – filter plate of gnathobase;
lat – lateral setae of exopodite;
mxp – maxillar process of limb I;
odl – outer distal lobe of limb I;
pep – preepipodite;
pos – posterior setae;
sdl – inner subdistal lobe of limb I.

## RESULTS

### Phylogenetics and phylogeography

We analyzed 118 specimens from 32 populations belonging to the *B. deitersi* group. The specimens originated mainly from Eurasia (Fig. 1), but a single population from North America and a single population from South America were also analyzed (Table S1).

We obtained 58 original sequences of *16S*, 15 sequences of *COI*, 43 sequences of *18S*, and 75 sequences of *28S*. Populations had a relatively high genetic polymorphism (Table 1). In contrast, the number of haplotypes was small. Each locus had a differing optimal substitution model (Table S2).

Three major clades of the *Bosminopsis deitersi* group were revealed from a tree based on the mitochondrial dataset (Fig. S1A). The first clade was *B. zernowi* – widely distributed in Eurasia and represented by two sub-clades (1 and 2). The second clade was *B. deitersi* distributed in the Americas, it is represented by two sub-clades: 1 in South America (*B. deitersi* s.str.) and 2 in North America. Both geographic subclades had modest support. Further study is necessary to examine the independent status of North American populations. A third clade (*Bosminopsis* sp.) was detected from a single population in Thailand.

The tree based on the nuclear dataset (*18S* + *28S*) had a similar topology to the mtDNA tree, but note that nuclear gene sequences were unavailable for *Bosminopsis* sp. from Thailand. The large clade of *B. zernowi* was again subdivided into two subclades (1 and 2) with low support. There were some conflicts between mitochondrial and nuclear sequences. Some specimens from the mitochondrial sub-clade 1 belonged to the nuclear sub-clade 2 (they are marked by asterisks in Figs. S1A–S1B). As support for both mitochondrial and nuclear subclades was low, we do not discuss this below. The *18S* locus was almost identical in all populations, suggesting the locus is most informative at the genus level. The *28S* locus demonstrated substantial variability in the D1 and D2 variable domains and appeared to contain information for taxa within the genus. Based on the neutrality tests and coalescent simulations in DnaSP v.6, we concluded that the most probable demographic model was a "bottleneck" model reflecting historical processes.

The final tree based on combined mitochondrial and nuclear datasets (Fig. S1C) was fully congruent with the mitochondrial tree–major clades were well-supported. No conflicts were found for ML and BI (with unlinked data) trees among genes or with the consensus tree.

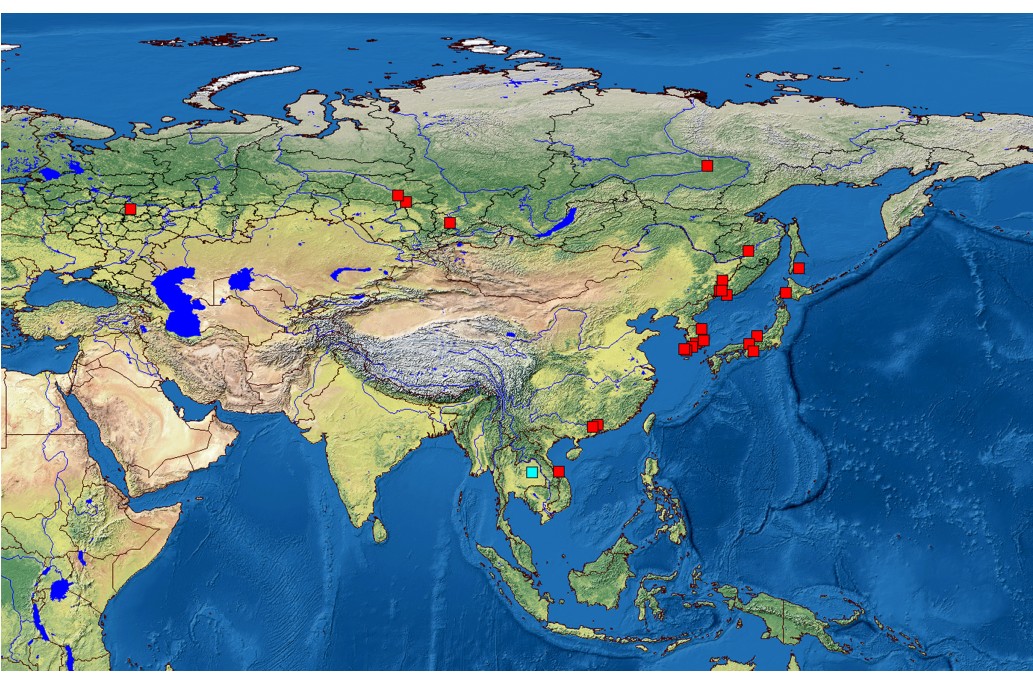

**Figure 1 Distribution of studied populations of the *Bosminopsis deitersi* group belonging to two major phylogroups:** *B. zernowi* **(red rectangles) and** *Bosminopsis* **sp. (blue rectangle).** Visualisation of the localities was made in free software DIVA-GIS7.5.0 ( https://www.diva-gis.org ) using Open Access spatial GIS data from http://www.naturalearthdata.com as the layers.

**Table 1 Genes, primers and annealing temperatures used in this study of the *Bosminopsis deitersi* species complex.**

| Gene | Primers | Sequence 5′-3′ | Temp. (°C) | Amplicon length | References |
|------|---------|----------------|-----------|-----------------|------------|
| COI | COI_Bosm_F | TGTAACAGCTCACGCATTTG | 50–54 | 415 | This paper |
| | COI_Bosm_R | ACCTGCTAGAACGGGAAGAC | 50–54 | | This paper |
| 16S | 16S-ar5 | CGCCTGTTTATC AA AACAT | 46–57 | 500–506 | (*Chapco, Kuperus & Litzenberger, 1999*) |
| | 16S-br3 | CCGGTCTGAACTCAGATCACGT | 46–57 | | (*Chapco, Kuperus & Litzenberger, 1999*) |
| 18S | 18a1 | CCTAYCTGGTTGATCCTGCCAGT | 52–59 | 562 | (*von Reumont et al., 2009*) |
| | 700R | CGCGGCTGCTGGCACCAGAC | 52–59 | | (*von Reumont et al., 2009*) |
| 28S | D1a | CCCSCGTAAYTTAAGCATAT | 48–50 | 370–373 | (*von Reumont et al., 2009*) |
| | D2b2 | CGTACTATTGAACTCTCTCTT | 48–50 | | (*von Reumont et al., 2009*) |

The results of the phylogenetic reconstructions suggested a deep demographic subdivision of the *B. deitersi* group. The tests of neutrality were consistent with such a division (Fu's Fs<=0 with Tajima D>>0). The most probable demographic process in this group evolution was an expansion with a strong founder effect resulting in strong differentiation between populations. We further explored the genetic diversity within each group and addressed the taxonomic uncertainty within these lineages.

For cybertaxonomic taxon delimitations (Fig. 2), both bGMYC and mPTP (for both mitochondrial and nuclear genes) suggested a deep divergence within the *B. deitersi* group.

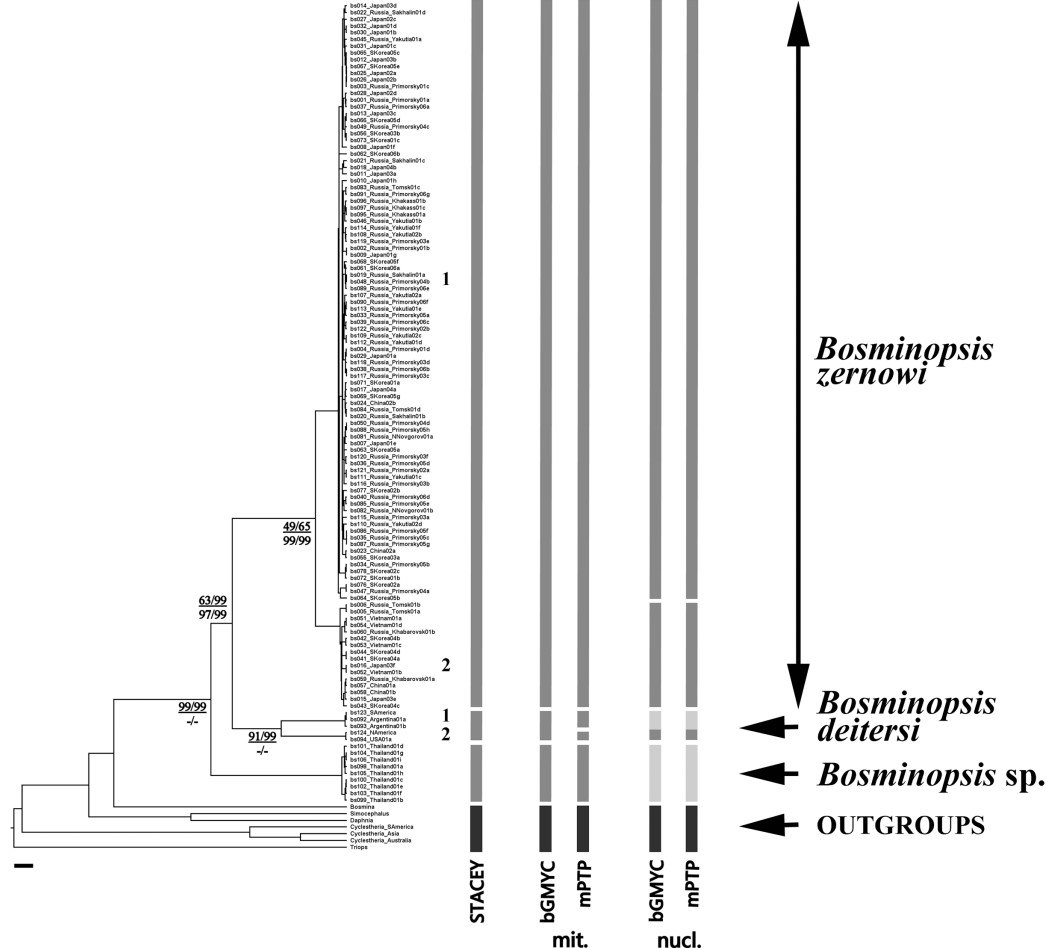

**Figure 2 BI multi-locus tree based on the *COI* + *16S* + *18S* + *28S* sequences, with a summary of results of the cybertaxonomic species delimitation by different methods.** Analyses referring are based on mitochondrial (mit.), nuclear (nuc.) and multi-locus datasets (STACEY). Node supports are: UFboot2 (ML) and posterior probabilities (BI), in percent for mitochondrial genes in the numerator and nuclear genes in the denominator. Dashes indicate branches that were not supported by a method.

All approaches suggested an independent status of *B. zernowi*, *B. deitersi* and *Bosminopsis* sp. from Thailand. Only the nuclear tree suggested a "Far Eastern" sub-clade based on the information in the hypervariable domains D1 and D2. There was some evidence that North American and South American populations of *B. deitersi* form independent species. More sampling of North American populations and loci is warranted to test this hypothesis further. Compared to bGMYC and mPTP, STACEY suggested significantly more taxa. However, increased splitting is expected (compared to morphological evidence) with STACEY (*Jones, 2017*; *Vitecek et al., 2017*).

To estimate divergences among selected OTUs, we calculated "simple" uncorrected *p*-distances for the best sampled locus, *16S* (Table 2). *Bosmina* was the outgroup. Distances among outgroups are ca. two times greater than the maximum distances within the *B. deitersi* groups. Groups "Eurasia", "Thailand", and "Americas" are well-differentiated,

**Table 2 Metrics of genetic diversity from mitochondrial and nuclear loci in *Bosminopsis*.**

| Loci | N | n | S | h | Hd | Pi | k | Fs | D | DemMod |
|---|---|---|---|---|---|---|---|---|---|---|
| COI (mit.) | 16 | 415 | 100 | 2 | 0.125 | 0.038 | 12.5 | 17.06 | −2.52* | PB (4.8) |
| 16S (mit.) | 60 | 500 | 99 | 8 | 0.647 | 0.044 | 23.9 | 26.70 | −0.04 | PD (4.9) |
| 18S (nucl.) | 43 | 562 | 6 | 5 | 0.221 | 0.001 | 0.36 | −3.02 | −2.11* | n/d |
| 28S (nucl.) | 75 | 377 | 50 | 5 | 0.392 | 0.039 | 14.3 | 26.25 | 0.99 | PB (0.01) |

**Note:**
N – number of sequences; n – total number of sites (excluding sites with gaps or missing data); S – number of segregating (polymorphic) sites; Hd – haplotype diversity; h – number of haplotypes; Pi – nucleotide diversity per site; k – average number of nucleotide differences; Fs – Fu's neutrality statistic (*Fu, 1997*) ; D – Tajima's D neutrality test (*Tajima, 1989*), the star-sign is indicated statistical significance *P* < 0.05; DemMod – most likely demographic model by DnaSP v.6 coalescent simulation (PB – population bottleneck, PD – population decline, n/d – not defined) based on Theta-W, probability P(Sim<Obs) is indicated as a percentage in brackets.

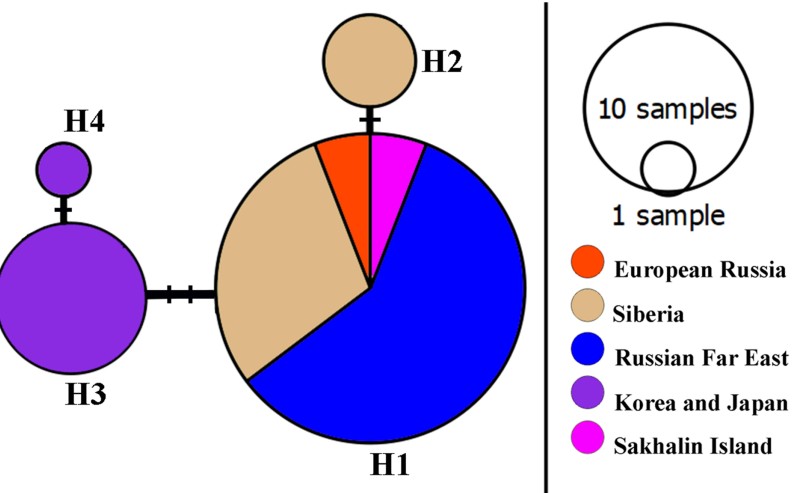

**Figure 3 A haplotype *16S* network for *Bosminopsis zernowi*.**

while differences between two sub-clades of Eurasia are less than 0.5%. The two subclades may result from moderately separated mitochondrial lineages, which are common in cladocerans (*Bekker et al., 2018*; *Kotov et al., 2021*). Again, North and South American populations may be separate species, but more sequences are needed to test this hypothesis.

A network of *16S* mitochondrial haplotypes (Fig. 3) revealed that all populations from Northern Eurasia belonged to only four haplotypes (Fig. 2): haplotype H1 included 73% of studied specimens and distributed from the Volga basin in European Russia to Pacific coast including Sakhalin Island (but not in Korea and Japan). H2 haplotype was endemic to the Yenisey Basin in Eastern Siberia and seemed to be a derivate of H1. Another well-represented haplotype (H3) differed by two substitutions from H1and was associated with a rare haplotype H4. H3 and H4 were detected only in Japan and Korea. Overall, haplotypic differentiation within groups was weak.

The Maximum Likelihood tests (Table 3) suggested that the hypothesis of molecular clocks is not rejected for each locus tested. In general, the topology of the tree constructed for the molecular clock calculations (Fig. 4A) is congruent with the multilocus tree

**Table 3 Estimates of evolutionary divergence over sequence pairs between groups.** The number of base differences per site from averaging over all sequence pairs between groups are shown. Standard error estimates are shown above the diagonal and were obtained by a bootstrap procedure (100 replicates). This analysis involved 61 nucleotide *16S* sequences. All ambiguous positions were removed for each sequence pair (pairwise deletion option). There was a total of 507 positions in the final dataset. Evolutionary analyses were conducted in MEGA-X (*Kumar et al., 2018*).

| Groups | 1 | 2 | 3 | 4 | 5 | 6 |
|---|---|---|---|---|---|---|
| Eurasia | | 0.003 | 0.014 | 0.014 | 0.015 | 0.020 |
| Far East | 0.004 | | 0.014 | 0.014 | 0.015 | 0.020 |
| North America | 0.122 | 0.122 | | 0.014 | 0.017 | 0.021 |
| South America | 0.116 | 0.112 | 0.128 | | 0.012 | 0.019 |
| Thailand | 0.114 | 0.110 | 0.137 | 0.084 | | 0.020 |
| OUT | 0.224 | 0.226 | 0.223 | 0.212 | 0.214 | |

**Table 4 Results of a molecular clock test using the Maximum Likelihood method.** Null hypothesis of equal evolutionary rate throughout the tree (at a 5% significance level).

| Locus | Substitution model in MEGA-X | TMC: with clock (lnL) | TMC: without clock (lnL) | Null |
|---|---|---|---|---|
| COI (mit.) 1st+2nd+3th | HKY+I | −1154.9 | −1152.1 | Not rejected |
| 16S (mit.) | GTR+G | −1588.4 | −1582.1 | Not rejected |
| 18S (nucl.) | JC | −1034.8 | −1033.1 | Not rejected |
| 28S (nucl.) | K2+G | −956.9 | −946.3 | Not rejected |

**Note:**
TMC – test for molecular clock in MEGA-X (*Kumar et al., 2018*). Models: JC – Jukes-Cantor, nst = 1 (*Jukes & Cantor, 1969*); K2 – Kimura 2-parameter, nst=2 (*Kimura, 1980*); HKY – Hasegawa-Kishino-Yano, nst = 2 (*Hasegawa, Kishino & Yano, 1985*); GTR – general time reversible, nst = 6 (*Rodriguez et al., 1990*). Non-uniformity of evolutionary rates among sites may be modeled by using a discrete Gamma distribution (+G) with 5 rate categories and by assuming that a certain fraction of sites are evolutionarily invariable (+I).

described above. Differences in the divergence pattern of the American (*B. deitersi*) and Thailand (*B.* sp.) clades may be explained by heterochrony in the appearance and fixation of the substitutions in mitochondrial and nuclear genomes (*Vawter & Brown, 1986*). Minor heterochrony is not an insurmountable obstacle for phylogenetic reconstructions (*Allio et al., 2017*).

"Simple" *p*-distances between *B. zernowi* and *B. deitersi* (based on the *COI* locus), gave a divergence time estimate of 17–30 MYA (*p* = 0.241). The estimate used an average divergence time for crustaceans of ca. 0.8–1.4 % per 1 Myr, which is similar to the age of divergence based on the coalescent model (Fig. 4A). Based on the time of divergence of the outgroup (*Bosmina*), we estimated the divergence of the *B. deitersi* group at around 200 MYA. Using RASP4, the taxa under consideration had an origin consistent with Laurasia (ancestral distribution range (C(ABCGEF)G), see Figs. 4B, 4C). However, note that Gondwanan populations from Africa, Australia, and India were not studied here. An alternative explanation is that the divergence of *Bosminopsis* sp. could be explained by its Gondwanan proto-range and subsequent colonization of Eurasia (i.e., due to India's continental drift). *Bosminopsis* sp. (ancestral range (G)) was separated in the Early

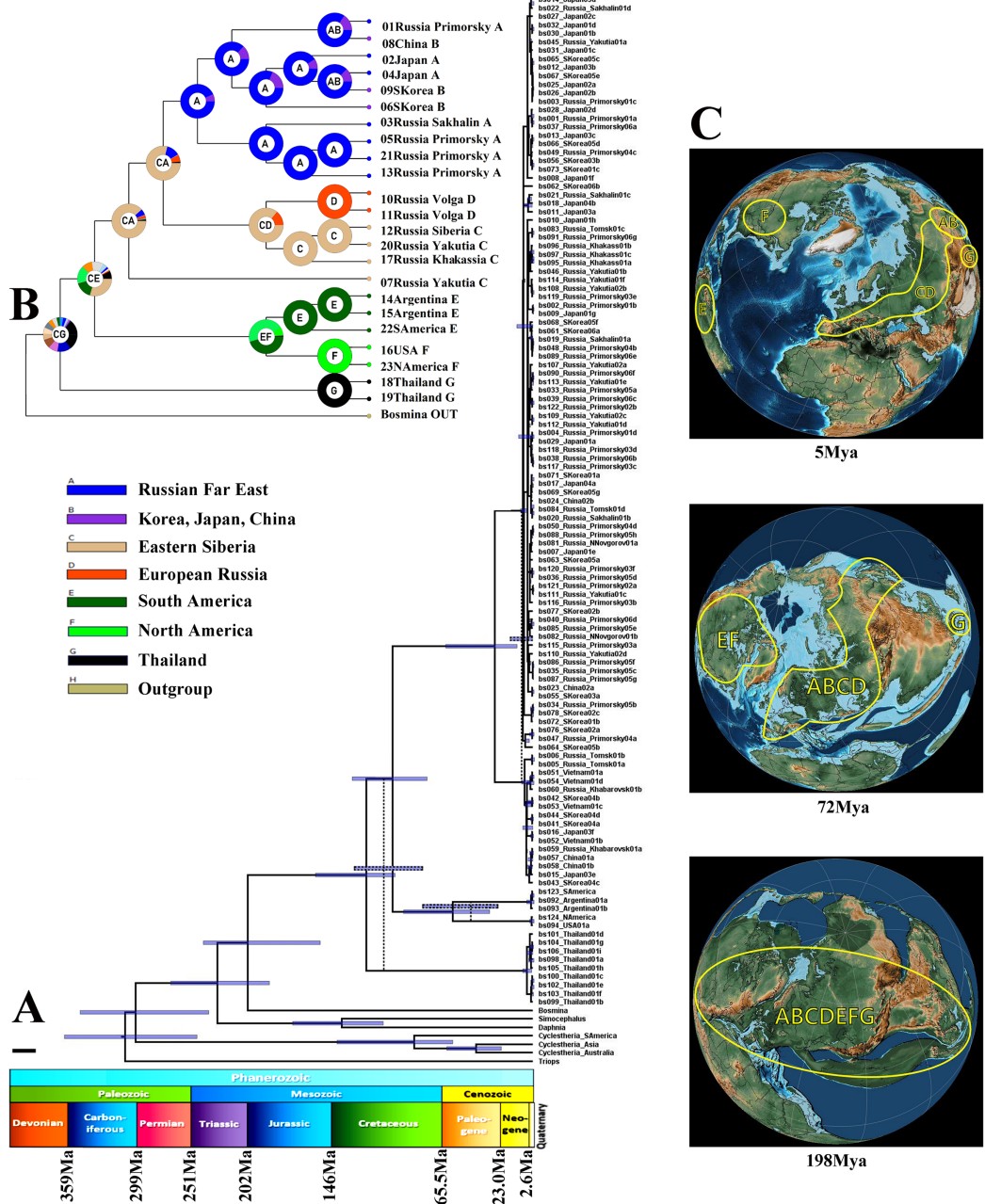

**Figure 4 Biogeographic history of *B. deitersi* group.** (A) – a possible phylogenetic tree for four loci based on the strict molecular clock, speciation by Yule process. Alternative topology of mitochondrial tree is represented by dotted line. Stratigraphic chart according to the International Commission on Stratigraphy (https://stratigraphy.org/chart). (B) – a proposed biogeographic history of the *B. deitersi* group on the consensus mitochondrial tree combined with the result of DIVALAKE+J model. Only tree topology is represented. Pie charts in each node demonstrate probabilities of alternative ancestral ranges; the most probable range is marked by the letter in the center. (C) – Possible ancestral ranges on palaeo-maps are represented at: 198 MYA, 72 MYA and 5 MYA. The maps are from the PalaeoAtlas for GPlates free software under GNU – General Public License (GPL) Ver. 2 (http://www.gnu.org/licenses/old-licenses/gpl-2.0.html).

Cretaceous from a Euro-Asian-American population of the group (CE(AB)GF)). Subsequent history was probably related to the disruption of Laurasia into Eurasian (CA (BCG)) and American (EF) groups of populations also in Cretaceous. Separation of North (F) and South American (E) populations had no ready explanation–it may be associated with the Gondwana-Laurasia split in the Cretaceous or a significantly more recent dispersal event (Neogene). A strong founder effect could then explain the genetic differences between Neogene populations of the two continents. In any event, the divergence of the entire *Bosminopsis* group is likely very ancient (at least the Early Cretaceous) and potentially affected by the split of proto-continents.

## Morphological analysis

**Order Anomopoda Sars, 1865**
**Family Bosminidae Baird, 1845**
**Genus *Bosminopsis* Richard, 1895**
**Short diagnosis.** Dorsal head pores absent in adults. Basal spine on postabdominal claw very large, as large as claw itself. Antennae I in females with proximal parts fused. Both exopod and endopod of antenna II three-segmented, antennal formula 0-0-3/1-1-3. Five pairs of thoracic limbs.

**Checklist of the formal taxa in the genus *Bosminopsis***

1. *Bosminopsis deitersi* Richard, 1895 – valid species.

2. *Bosminopsis zernowi* Linko, 1901 – valid species.

3. *Bosminella Anisitsi* Daday, 1903 – junior synonym of *B. deitersi*.

4. *Bosminopsis ishikawai* Klocke, 1903 – junior synonym of B. *zernowi*.

5. *Bosminella Anisitsi* var. *africana* Daday, 1908 – status must be checked, it could be a valid species.

6. *Bosminopsis deitersi* var. *typica* Burckhardt, 1909 – junior synonym of *B. deitersi*.

7. *Bosminopsis deitersi birgei* Burckhardt, 1924 – valid species.

8. *Bosminopsis deitersi brehmi* Burckhardt, 1924 – junior synonym of *B. africana*.

9. *Bosminopsis deitersi klockei* Burckhardt, 1909 – junior synonym of B. *zernowi*.

10. *Bosminopsis deirestsi pernodi* Burckhardt, 1924 – possible junior synonym of B. *zernowi*.

11. *Bosminopsis deirestsi schroeteri* Burckhardt, 1924 – junior synonym of *B. zernowi*.

12. *Bosminopsis stingelini* Burckhardt, 1924 – junior synonym of *B. deitersi*.

13. *Bosminopsis deitersi* var. *africana* Rahm, 1956 – junior homonym of *B. africana*.

14. *Bosminopsis negrensis* Brandorff, 1976 – valid species, endemic of Brazil.

15. *Bosminopsis devendrari* Rane, 1984 – *species inquirenda*, it could be a valid taxon from SE Asia.

16. *Bosminopsis macaguensis* Rey & Vasquez, 1986 – junior synonym of *B. deitersi* (see Kotov, 1997b).

17. *Bosminopsis brandorffi* *Rey & Vasquez, 1989* – valid species, endemic of Brazil.

Unavalable name:

18. *Bosminopsis granulata* Daday – unpublished taxon name for Indian populations; slides of E. Daday labeled by this name are kept in the Collectio Dadayana of the Hungarian Natural History Museum, Budapest, Hungary.

### *Bosminopsis deitersi* group

**Diagnosis.** Valve with a single mucro or several mucro-like spines at postero-ventral valve portion. No postero-dorsal spine (caudal needle) on carapace. Basis of postabdominal setae not inflated.

**Comments.** Among 17 available taxa listed above, 15 belong to the *B. deitersi* group. Only two valid species are not members of the *B. deitersi* group, both are andemics of Amazonia (*B. negrensis* and *B. brandorffi*, numbers 14 and 17 in our checklist). Most taxa of the *B. deitersi* group were poorly described. Here we try to start the revision of the group, redescribing *B. deitersi* s.str. and *B. zernowi*. We do not have adequate material (i.e. populations with males and ephippial females) for revision of North American, SE Asian, African, and Australian taxa.

### *Bosminopsis deitersi* *Richard, 1895* s.str.
Figures 5A–E, 6–9

*Bosminopsis deitersi* *Richard, 1895*. Richard, 1895, p. 96–98, figs 1–4; *Richard, 1897*, p. 283–286, figs 28–31; *Stingelin, 1904*, p. 584–586, Pl. 20: figs 7–10; *Burckhardt, 1909*, p. 251; *Burckhardt, 1924*, p. 221–228; *Rey & Vasquez, 1986*, p. 222–225, Pl. 2: figs 1–16; *Kotov, 1997a*, p. 26–29, figs 1–2; Kotov, 1997b, p. 6–26, figs 1–13; *Kotov & Ferrari, 2010*, p. 51.
*Bosminopsis deitersi* var. *typica* n.n. in *Burckhardt, 1909*, p. 251.
*Bosminopsis stingelini* *Burckhardt, 1909*. *Burckhardt, 1909*, p. 251, text-figure: A-B; *Burckhardt, 1924*, p. 228–229, figs 2, 8.
*Bosminopsis macaguensis* *Rey & Vasquez, 1986*. *Rey & Vasquez, 1986*, p. 220–222, Pl. 1: figs 1–18.
*Bosminella Anisitsi* *Daday, 1903*. *Daday, 1903*, p. 594–597, figs 1–3; *Daday, 1905*, p. 199–200, Pl. 13: figs 1–5.

**Type locality.** "… l'eau douce à La Plata (Buenos-Ayres)" (*Richard, 1895*), Argentina.

**Type material.** Lost, absent in the Collection of Jules Richard at the National Museum of Natural History, U.S.A. (*Kotov & Ferrari, 2010*).

**Material studied here.** See Table S1.

**Short diagnosis.** Body of large adult parhenogenetic female subovoid, in younger adults more elongated, with a short postero-dorsal spine, but a caudal needle absent. Reticulation

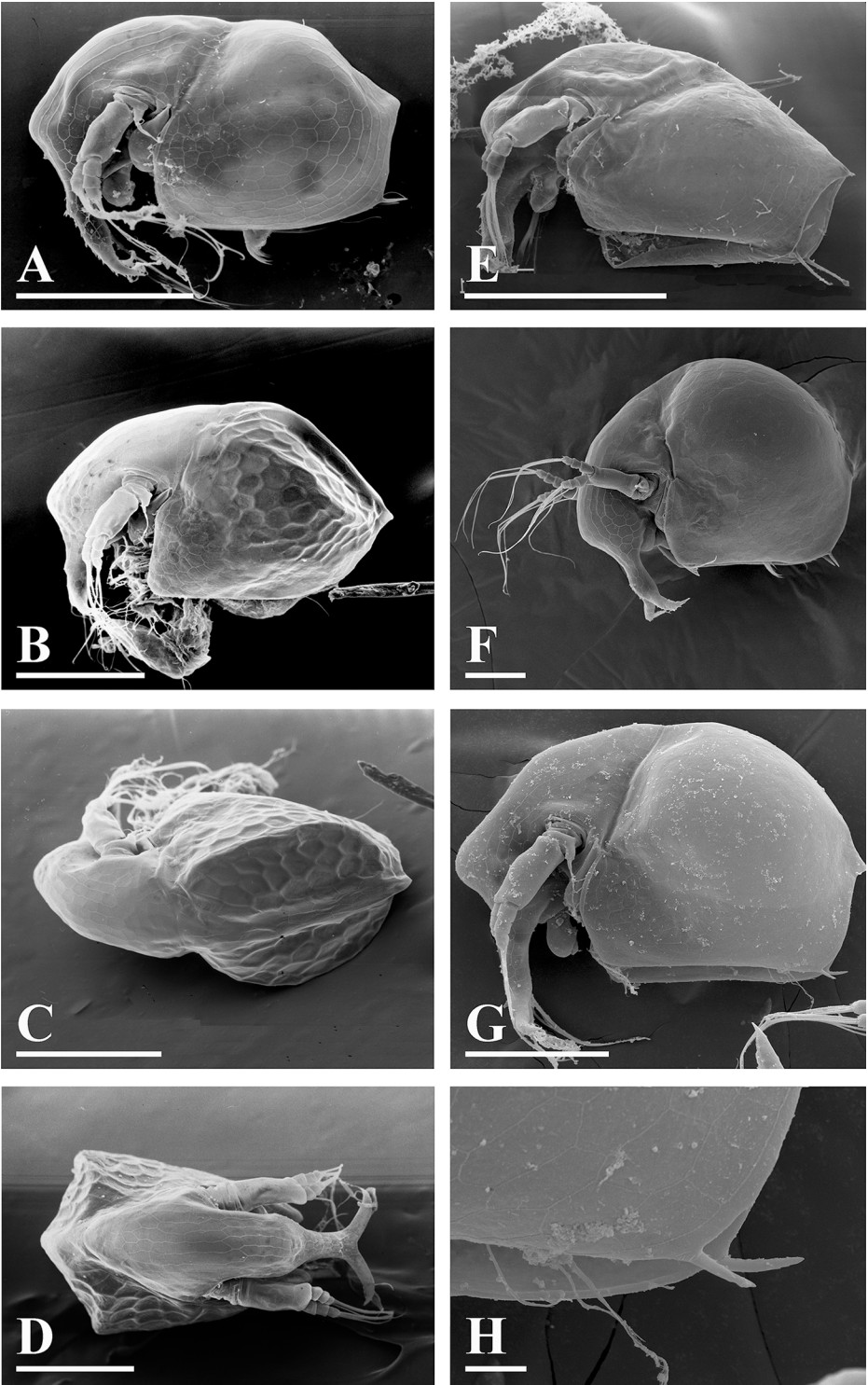

**Figure 5** *Bosminopsis deitersi Richard, 1895* **from Brazil (A–E) and** *Bosminopsis* **sp. from Bung Pueng, Kalasin Province (F) and Lake Bueng Khong Long, Nong Khai Province (G–H) in Thailand (F–H).** A, Adult parthenogenetic female from Rio Xingu. (B–D), Ephippial female from Lago do Castanho, lateral, dorsal and anterior view. (E), Juvenile female from Rio Tapajos. (F). Large adult parthenogenetic female. (G–H), Juvenile female and its mucro. Scale bars: A–G = 0.1 mm, H = 0.01 mm.               

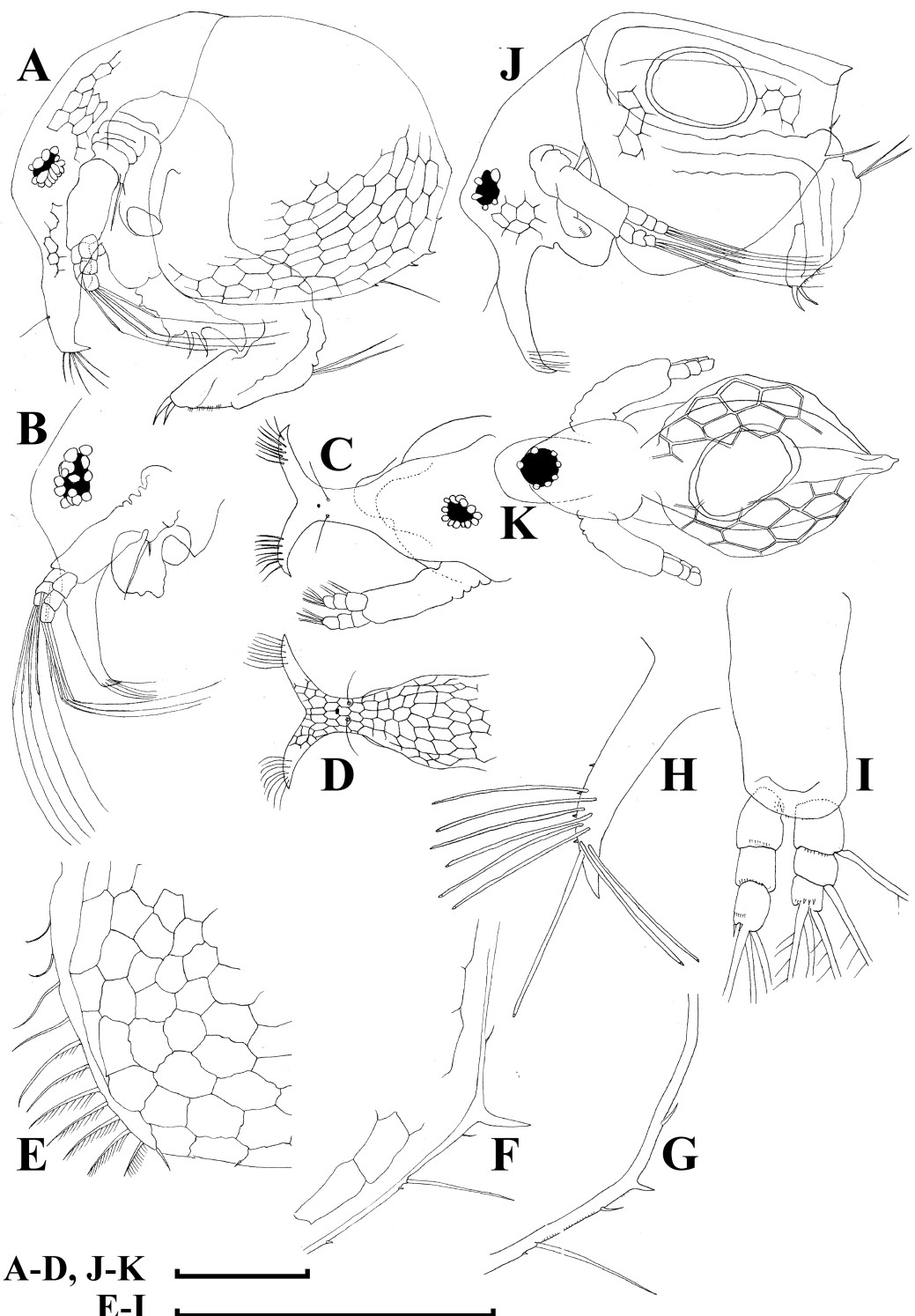

**Figure 6** *Bosminopsis deitersi Richard, 1895*, **parthenogenetic (A–I) and ephippial (J–K) females from Lago do Castanho and Lago Cristalino, both in Amazonas, Brazil.** (A), Adult parthenogenetic female, lateral view. (B), Its head, lateral view. (C–D), Head, anterior view. (E), Antero-ventral portion of valve. (F–G), Posteroventral portion of valve. (H), Antenna I. (I), Antenna II. (J), Mature ephippial female, lateral view. (K), Mature ephippial female, dorsal view. Scale bars = 0.1 mm.

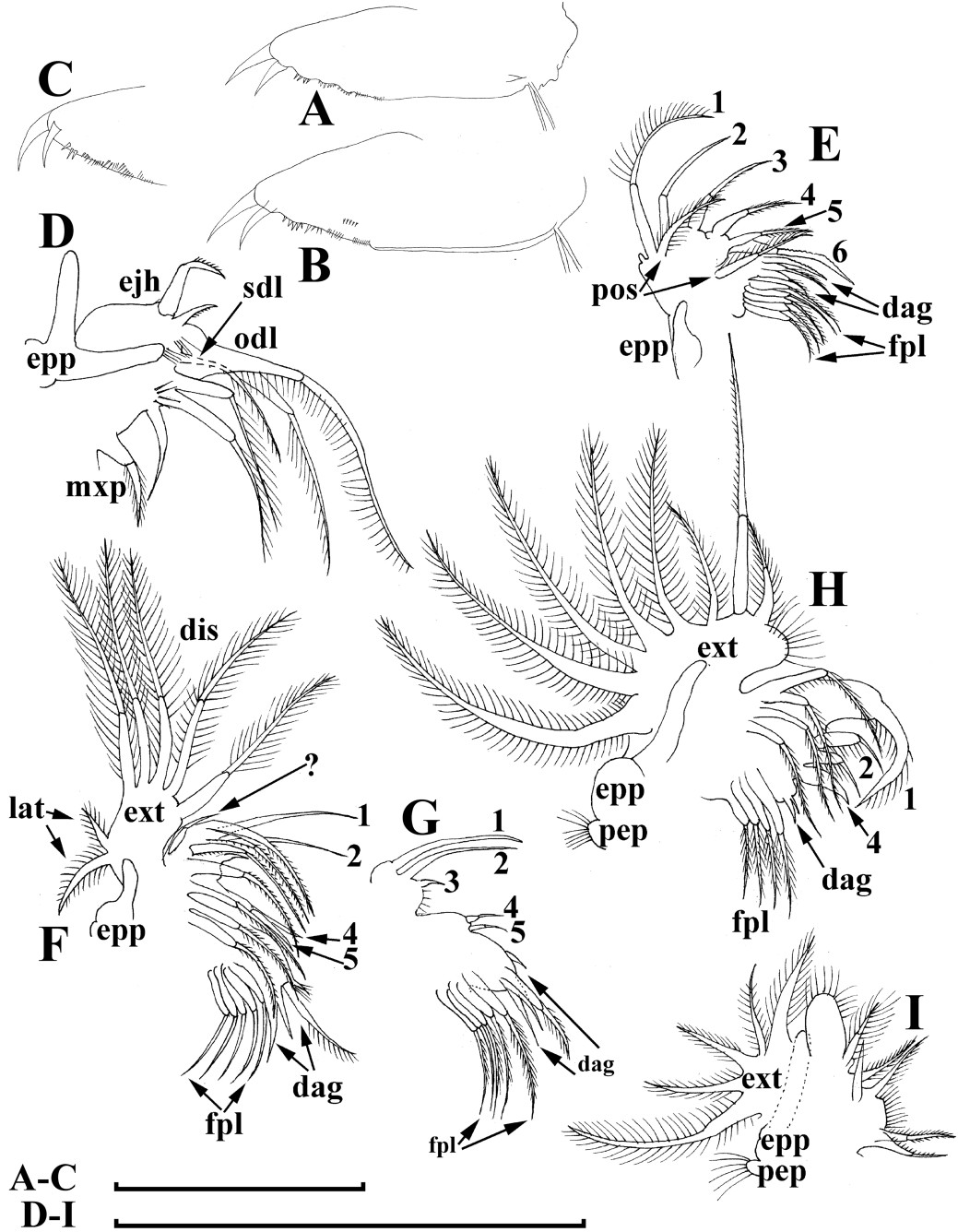

**Figure 7** *Bosminopsis deitersi Richard, 1895*, parthenogenetic female from Lago do Castanho, Amazonas, Brazil. (A–C), Postabdomen, lateral view. (D), Limb I. (E), Limb II. (F), Limb III. (G), Gnathobase of limb III. (H), Limb IV. (I), Limb V. Scale bar = 0.1 mm.

well-expressed on valves and head. Valve with a single short mucro at postero-ventral valve portion, or it is completely reduced. Postabdomen without inflated basis of postabdominal setae. Limb I with epipodite having two finger-like projections. Juvenile female with a long postero-ventral mucro, supplied by minute denticles. Free and fused

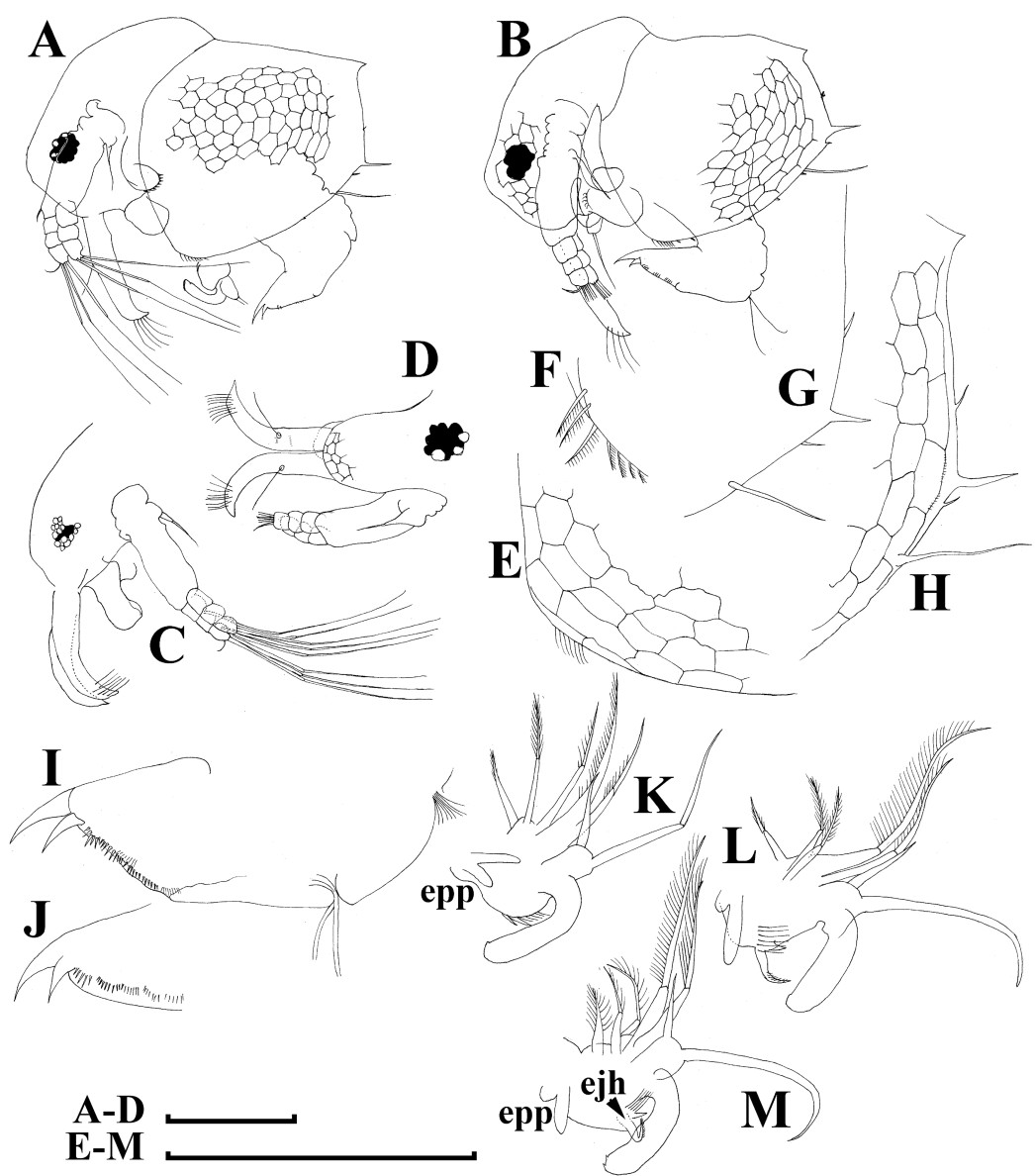

**Figure 8** *Bosminopsis deitersi Richard, 1895*, **juvenile male from Lago do Castanho (Amazonas, Brazil).** (A–B), Lateral view. (C), Head, lateral view. (D), Head, anterior view. (E–F), Antero-ventral portion of valve. (G–H), Postero-ventral portion of valve. (I–J), Postabdomen. (K–M), Limb I. Scale bars = 0.1 mm.

parts of antennae I, mucro, rostrum, ventral valve edge, base of caudal spine covered with small spinules. Ephippial female with egg chamber sculpture represented by large polygons. A strong medial keel on dorsum, strong paired lateral keels well distinguishable from the dorsal view. Adult male with dorsal contour of head humped, head large, with a smooth rostrum and expressed ocular dome, a short mucro always present postero-ventrally. Postabdominal claw bears a basal spine comparable in size with the latter. Antenna I free, remarkably curved distally. A relatively long (somewhat longer than exopod itself), curved at tip additional male seta on endopod apical segment in position of

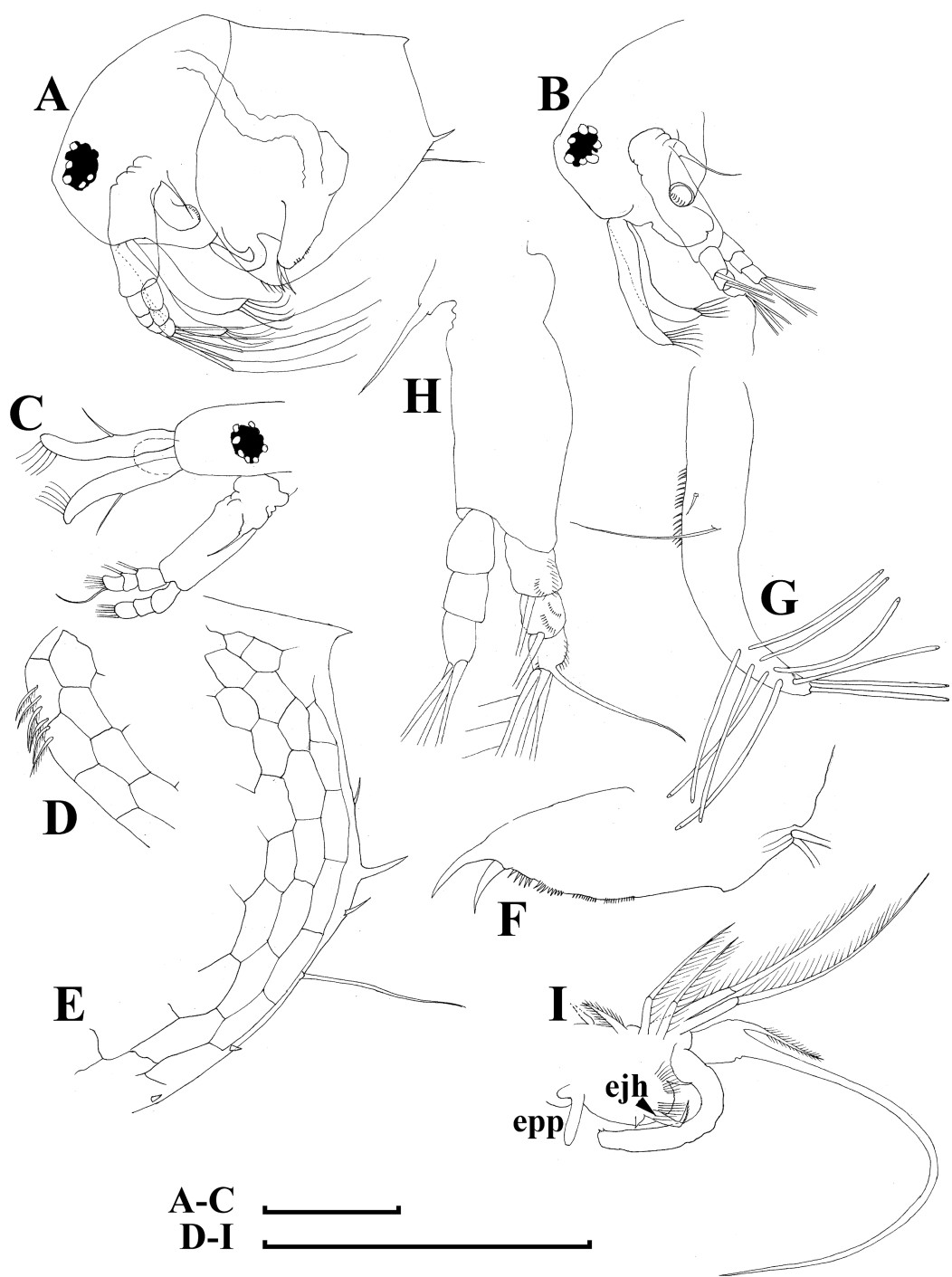

**Figure 9** *Bosminopsis deitersi Richard, 1895*, **adult male from Lago do Castanho, Amazonas, Brazil.**
(A–B), Lateral view. (C), head, anterior view. (D), Antero-ventral portion of valve. (E), Postero-ventral
portion of valve. (F), Postabdomen. (G), Antenna I. (H), Antenna II. (I), Limb I. Scale bars = 0.1 mm.

a rudimentary spine in female. Limb I a copulatory hook relatively large and regularly thick, its tip blunt. Size 0.17–0.41.

## Redescription

**Adult parthenogenetic female.** Body short and almost round in lateral view (body height/length ratio about 0.65–0.69), dorsal margin regularly curved from base of antenna I to posterodorsal angle (Figs. 5A, 6A). Reticulation prominent, both on head and on valves. Posterior margin straight, with height about half of total body height, postero-dorsal angle expressed. Head in lateral view with a low ocular dome (Fig. 6B), body contour between head and proboscis rostral part (fused bases of antennae I) depressed. Frontal head pore ovoid, located almost in the middle of rostral part, somewhat anteriorly to level of frontal sensory setae (Figs. 6C–6D). Lateral and dorsal head pores absent in adults. Compound eye of moderate size. Ocellus absent. Labrum as a fleshy appendage, its anterior contour convex. Antero-ventral portion of valves with setulated setae (Fig. 6E), ventral margin slightly convex, with a series of spinules, a long seta (*seta Kurzi*) and a rudimentary mucro at poster-ventral angle (Figs. 6F–6G). Postabdomen compressed laterally, slightly and regularly narrowing distally, without inflated basis of postabdominal setae (Figs. 7A–7C). Preanal margin long, straight to slightly concave, without setules. Anal margin straight, preanal angle expressed, but postanal angle absent. Anal and postanal portion with small denticles and as a postabdominal claw terminally supplied with a strong basal spine, almost as large as claw, both claw and basal spine slightly curved. Postabdominal seta bisegmented, shorter than postabdomen.

Proximal portions of two antennae I fused together and with rostrum, both lateral portions directed downwards and slightly curved laterally (Figs. 6C–6D, 6H). Antennular sensory setae located on fused portion of antennas I. Distal portions with nine aesthetascs subequal in size. Antenna II (Fig. 6I) with a coxal portion bearing a long seta and a short seta on a conical elevation, elongated basal segment and short three-segmented exopod and endopod, antennal formula: setae 0-0-3/1-1-3; spines 0-0-1/0-0-1, but apical spines greatly reduced in size. All apical and lateral (on endopod first and second segment) setae subequal in size, covered by fine setules.

Limb I large, its corm conically narrowing distally. Epipodite (Fig. 7D: epp) with two long finger-like projections. Outer distal lobe (Fig. 7D: odl) with two setae of different size, feathered by sparse, long, robust setules. Inner subdistal lobe (in terms of *Kotov, 1997a*) (Fig. 7D: isl) with a single seta, densely fringed by delicate setules. On inner limb edge, three soft setae. A bunch of long setules is located near these setae. Two robust ejector hooks (Fig. 7D: ejh) strongly different in size, armed with short denticles. The maxillar process (Fig. 7D: mxp), a derivative of gnathobase I, with a single long, densely setulated seta, at base of the limb.

Limb II relatively small, with epipodite supplied by a finger-like projection. Inner limb portion with an anterior row of 6 setae (homologs of "scrapers" of the chydorids, see *Fryer, 1968*) (Fig. 7E: 1–6) and disjuncted posterior row two setae (Fig. 7E: pos): a seta near gnathobase and another one near the proximal end of the limb. Gnathobase II with distal

armature (Fig. 7E: dag) of three setae of different armature. Filter plate consists of five long setulated setae (Fig. 7E: fpl).

Limb III with epipodite (Fig. 7F: epp) supplied with a finger-like projection. Exopodite rectangular, bearing two lateral (Fig. 7F: lat=6–7) and five (Fig. 7F: dis=1–5) distal setae, seta 1 shortest among distal setae. Each seta covered by long setules. Distal endite (in terms of *Kotov, 2013*) with three anterior setae (Figs. 7F, 7G: 1–3): setae 1 and 2 long; seta 3 especially short. Proximal endite with two small anterior setae (Figs. 7F, 7G: 4–5). Eight soft setae on posterior face of limb, plus a seta of unclear homology (Fig. 7F: ?). Distal armature of gnathobase (Fig. 7F: dag) with three setae and a small sensillum. Filter plate (Fig. 7F: fpl) with five setae of subequal size.

Limb IV with small ovoid setulated pre-epipodite (Fig. 7H: pep) and a finger-like epipodite (Fig. 7H: epp). Exopodite circular with eight soft setae (1–8), no subdivision into lateral and distal setae. The longest seta covered by fine stiff setulae, others with long setules. The distalmost portion of exopodite as a densely setulated flat lobe. Inner distal portion with four anterior setae (Fig. 7H: 1–4); among them distal most setae 1 especially thick. Four thin long setae on posterior limb face. Distal armature of gnathobase (Fig. 7H: dag) with two elements represented by a thin sensillae. Filter plate (Fig. 7H: fpl) with four setae subequal in size.

Limb V (Fig. 7I) with a small, ovoid setulated preepipodite and an epipodite supplied with a long finger-like projection. Exopodite with five soft setae (1–5) covered by long setules, seta 5 exceptionally long. The distalmost portion of limb as a densely setulated flat lobe, two soft setae near it, two setulated setae of subequal length near gnathobase. Filter plate with two long setae.

**Juvenile female.** Instar I has a dorsal head pore (*Kotov, 1997b*). Body elongated, head relatively high, elevating over valves, without a cervical incision (Fig. 5E). Carapace with a short posterior spine and a long postero-ventral mucro, supplied with minute denticles. Antenna I relatively longer than in female. Free and fused parts of antennules, mucro, rostrum, ventral valve edge, base of caudal spine covered with relatively small spinules.

**Ephippial female.** Only dorsal portion of valves modified as compared to parthenogenetic female (Figs. 5B–5D, 6J–6K). Ephippium yellowish, ovoid, not clearly demarked from ventral and lateral portions of valves. Egg chamber with a single egg, elongated, its sculpture represented by large polygons well visible under light microscope with very clearly, minute wrinkles and tubercles in each polygon. A strong medial keel on dorsum, strong paired lateral keels well distinguishable from the dorsal view. From the dorsal view, keels projected laterally out of body contour.

**Juvenile male.** Body elongated, with a clear dorsal depression posteriorly to head (Figs. 8A–8B). Head large, with ill-developed ocular dome (Figs. 8C–8D). Armature of antero-ventral valve portion (Fig. 8E) as in female. Mucro well-developed (Figs. 8G–8H). Postabdomen short, gonopores not visible (Figs. 8I–8J). Antennae I fused to rostrum, but their bases are not fused together (Fig. 9D). Limb I with a short, thick copulatory hook (Figs. 8K–8M).

**Adult male.** Shape significantly different from that in female, body short (body height/ length ratio about 0.65), dorsal contour of head humped, dorsal contour of carapace straight, valve anterior portion with few setae anteriorly, ventral margin convex, with setules as in female (Fig. 9A). Head large, with a smooth rostrum and expressed ocular dome, compound eye large (Figs. 9B–9C). Valve armature as in female (Fig. 9D), but a short mucro always present postero-ventrally (Fig. 9E). Postabdomen similar with that in female, its ventral margin slightly comvex, preanal margin slightly to moderately concave. Anal margin almost straight, postanal angle absent. Postabdominal claw bears a basal spine comparable in size with the latter (Fig. 9F), both claw and basal spine slightly and regularly bent.

Antenna I free, remarkably curved distally (Fig. 9G). Frontal sensory seta long, located at middle of antennular body, a short male seta somewhat anteriorly to that, several fields of short spinules located at antenna I anterior face. Long aesthetascs located subterminally, two of them are located on the tip of antenna I, the others located on its lateral surface in two rows. Antenna II with apical and lateral setae as in female. A relatively long (somewhat longer than exopod itself), curved at tip additional male seta on endopod apical segment in position of a rudimentary spine in female (Fig. 9H). Limb I with outer distal lobe bearing two setae strongly unequal in size, copulatory hook relatively large and regularly thick, its tip blunt, not expanded bearing small denticles (Fig. 9I).

**Size.** Females 0.17–0.45 mm, adult males 0.29–0.31 mm.

**Differential diagnosis.** *B. deitersi* differs from *B. zernowi* in (1) only a single mucro at postero-ventral valve angle in both females and males; (2) different proportions of setae in exopodite, inner limb portion and distal armature of gnathobase of limb III and on exopodite V; (3) male basal spine on postabdominal claw significantly shorter that the claw itself; (4) male antenna I strongly bent distally; and (5) additional seta on apical segment of male antenna II curved at tip. Morphological differences from other taxa revealed above genetically are not studied yet.

**Distribution and ecology.** Widely distributed in the Neotropical zone. Records from Mexico (*Elias-Gutierrez et al., 2008*) and Central America (*Collado, Fernando & Sephton, 1984*) need to be checked as they could belong to *B. deitersi* s.str. or the poorly described *B. birgei*.

Populations with a single mucro in juveniles are present on other continents (Figs. 5F–5H), but they belong to other taxa that need to be revised.

***Bosminopsis zernowi* Linko, 1901**
Figures 10–14

gen.? sp.? in *Zernov, 1901*, p. 34, Pl. 4: Fig. 27.
*Bosminopsis zernowi Linko, 1901*. *Linko, 1901*, p. 345–347, text-fig.; *Meissner, 1902*, p. 52; *Meissner, 1903*, p. 180–190, Plates 2–4; *Zykoff, 1906*, p. 22–24, text-fig.; *Burckhardt, 1909*, p. 251; *Burckhardt, 1924*, p. 229–230.

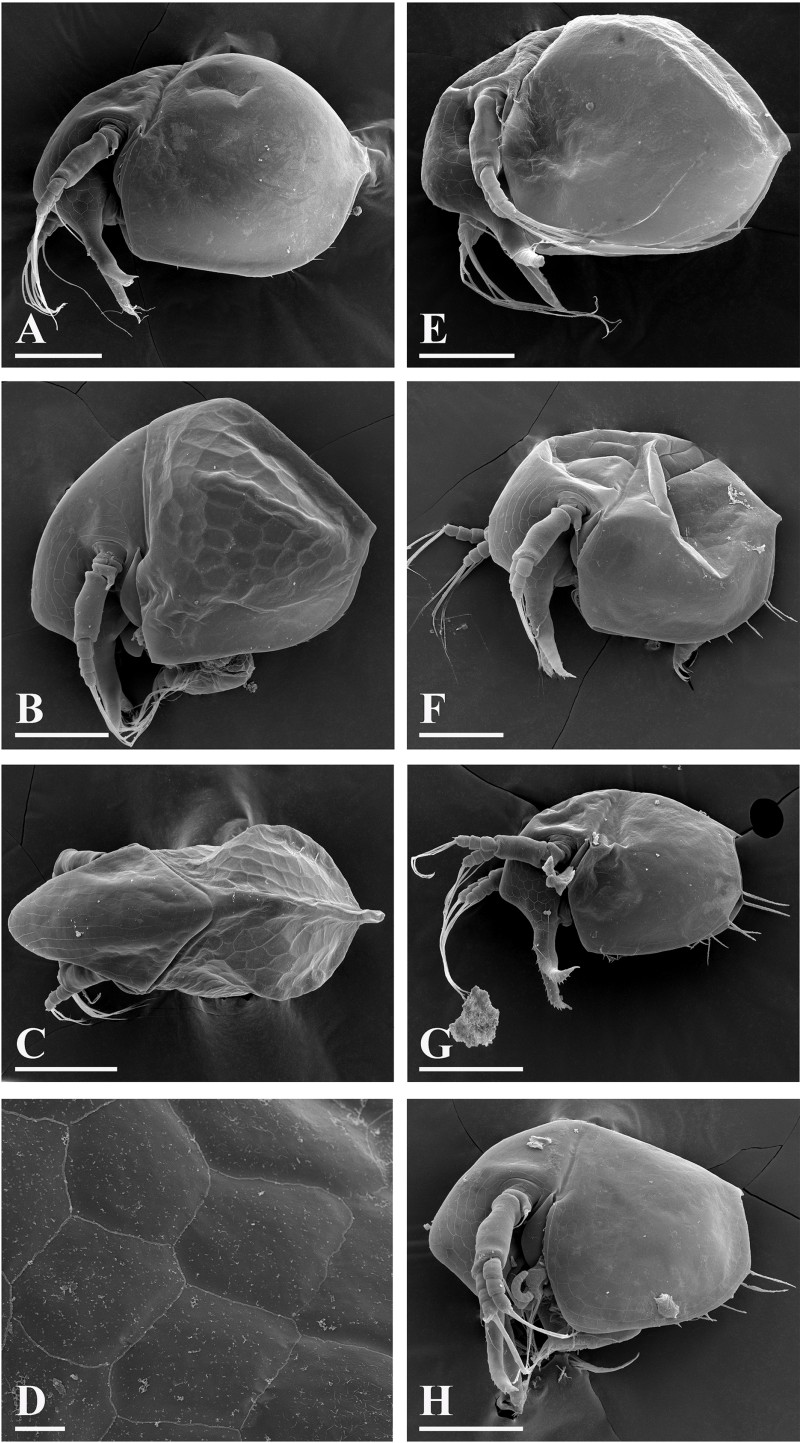

**Figure 10** *Bosminopsis zernowi Linko, 1900* from Sai-no-Kami Ike, Japan (A, C, G), Lake Ilinskoe, Primorsky Territory, Russia (B, D, F, H) and Lena River near Yakutsk, Yakutia Republic, Russia (E). (A), Adult parthenogenetic female. (B–D), Ephippial female in lateral and dorsal view and sculpture of ephippium. (E), Pre-ephippial female. (F–G), Juvenile female; (H), Juvenile male II. Scale bars: A–C, E–H = 0.1 mm, D = 0.01 mm.               

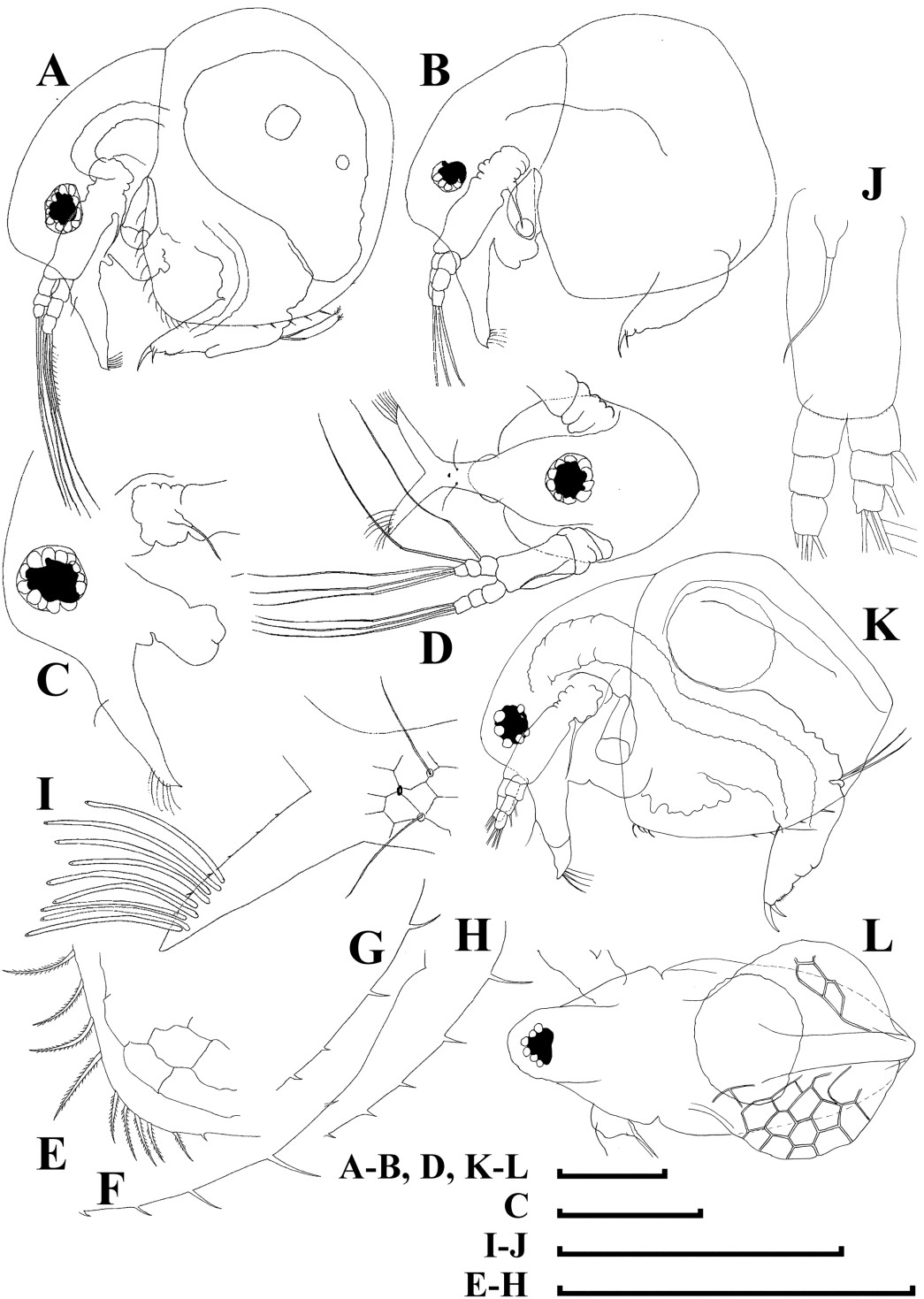

**Figure 11** *Bosminopsis zernowi Linko, 1901*, **large parthenogenetic females from Ivankovskoe Water Reservoir on Volga River, European Russia (A–J) and mature ephippial female from a tributary of Dnepr River, Ukraine (K–L).** (A–B), Lateral view. (C), Head, lateral view. (D), Head, anterior view. (E), Setae at antero-ventral valve portion. (F–H), Spines at postero-ventral valve margin. (I), Antenna I. (J), Antenna II. (K), Ephippial female, lateral view. (L), Its dorsal view. Scale bars = 0.1 mm.

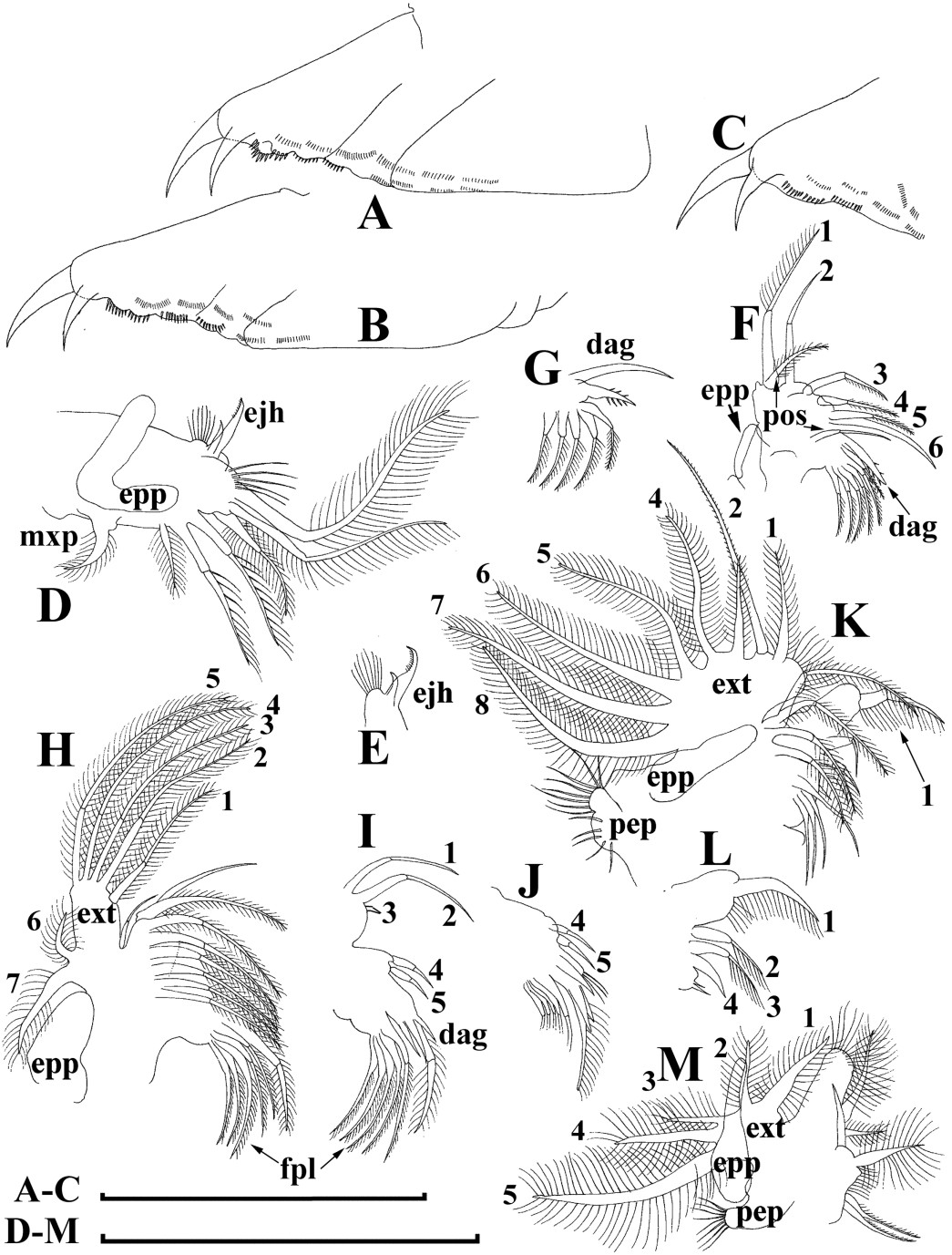

**Figure 12 *Bosminopsis zernowi Linko, 1901*, parthenogenetic female Ivankovskoe Water Reservoir on Volga River, Tver' Area, European Russia.** (A–C), Postabdomen. (D), Limb I. (E), Ejector hooks I. (F), Limb II. (G), Distal armature of its gnathobase. (H), Limb III. (I), Its inner-distal portion. (J), Granthobase III. (K), Limb IV. (L), Its inner-distal portion. (M), Limb V. Scale bar = 0.1 mm.

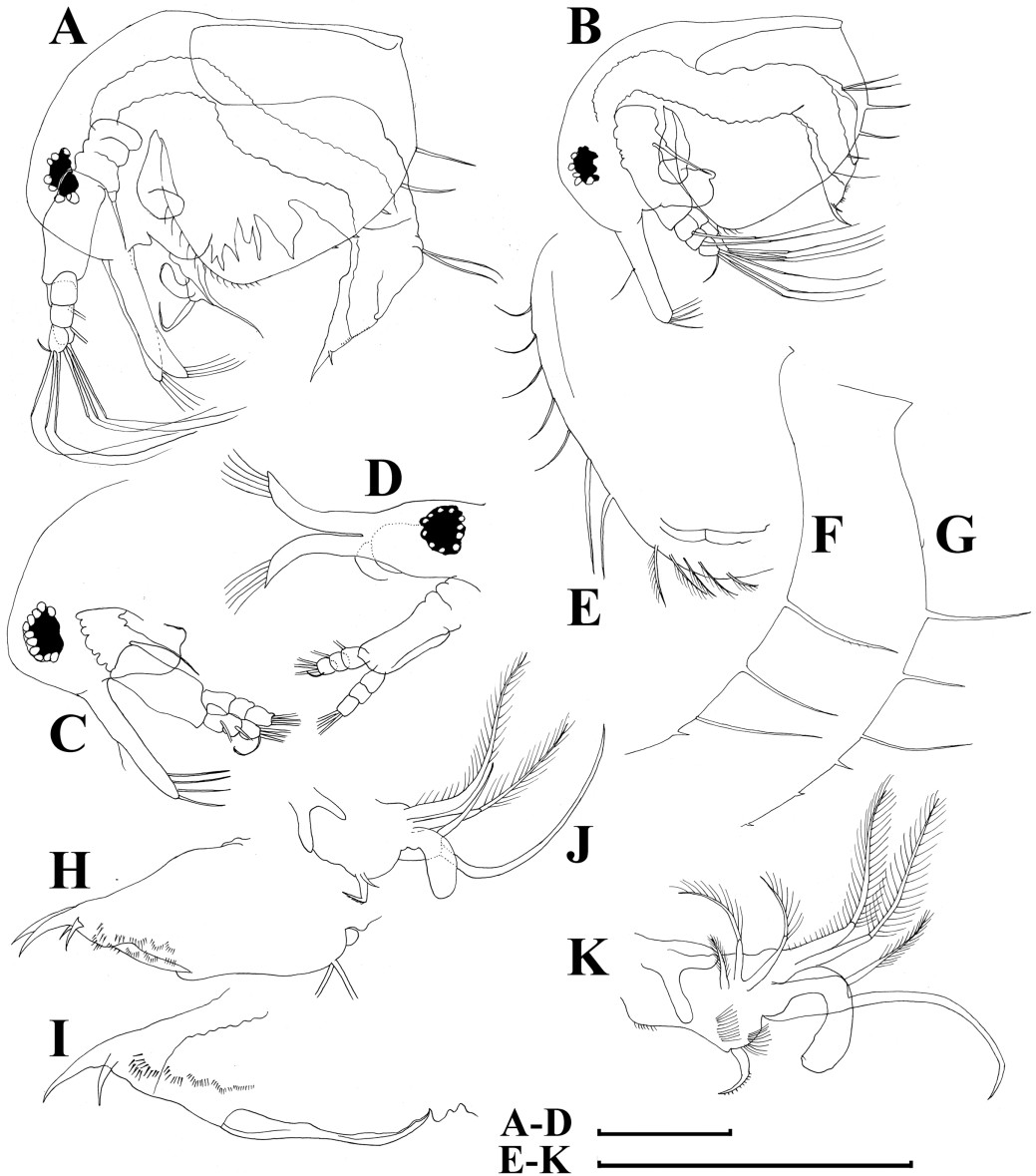

**Figure 13** *Bosminopsis zernowi Linko, 1901*, **juvenile male of instar II (A, C–G, I, K) and instar I (B, H, J) from Lake Livadijskoe, Primorski Territory, Far East of Russia.** (A–B), Lateral view. (C), Head, lateral view. (D), Its anterior view. (E), Antero-ventral valve portion. (F–G), Posterior portion of valve. (H–I), Postabdomen. (J–K), Limb I. Scale bars = 0.1 mm.

*Bosminopsis deitersi zernowi* in *Behning, 1941*, p. 190–191, Fig. 83; *Manujlova, 1964*, p. 265, Fig. 147 (1, 3).
*Bosminopsis deitersi* in *Krasnodebski, 1937*, p. 357–360, Pl. 12: Fig. 1; Smirnov, *1995*, p. 66, Fig. 58 (1–2); *Song & Mizuno, 1982*, p. 343, Fig. 2–3; *Yoon & Kim, 1987*, p. 194, Fig. 8e–g; *Kim, 1988*, p. 58, Fig. 40; *Lieder, 1996*, p. 29–31, Fig. 1a–c, 2a-f; *Tanaka, 2000*, p. 110, Fig.1–2; *Yoon*, 2010, p. 94–95, Fig. 49; *Jeong, Kotov & Lee, 2014*, p. 221; *Bledzki & Rybak, 2016*, p. 172.

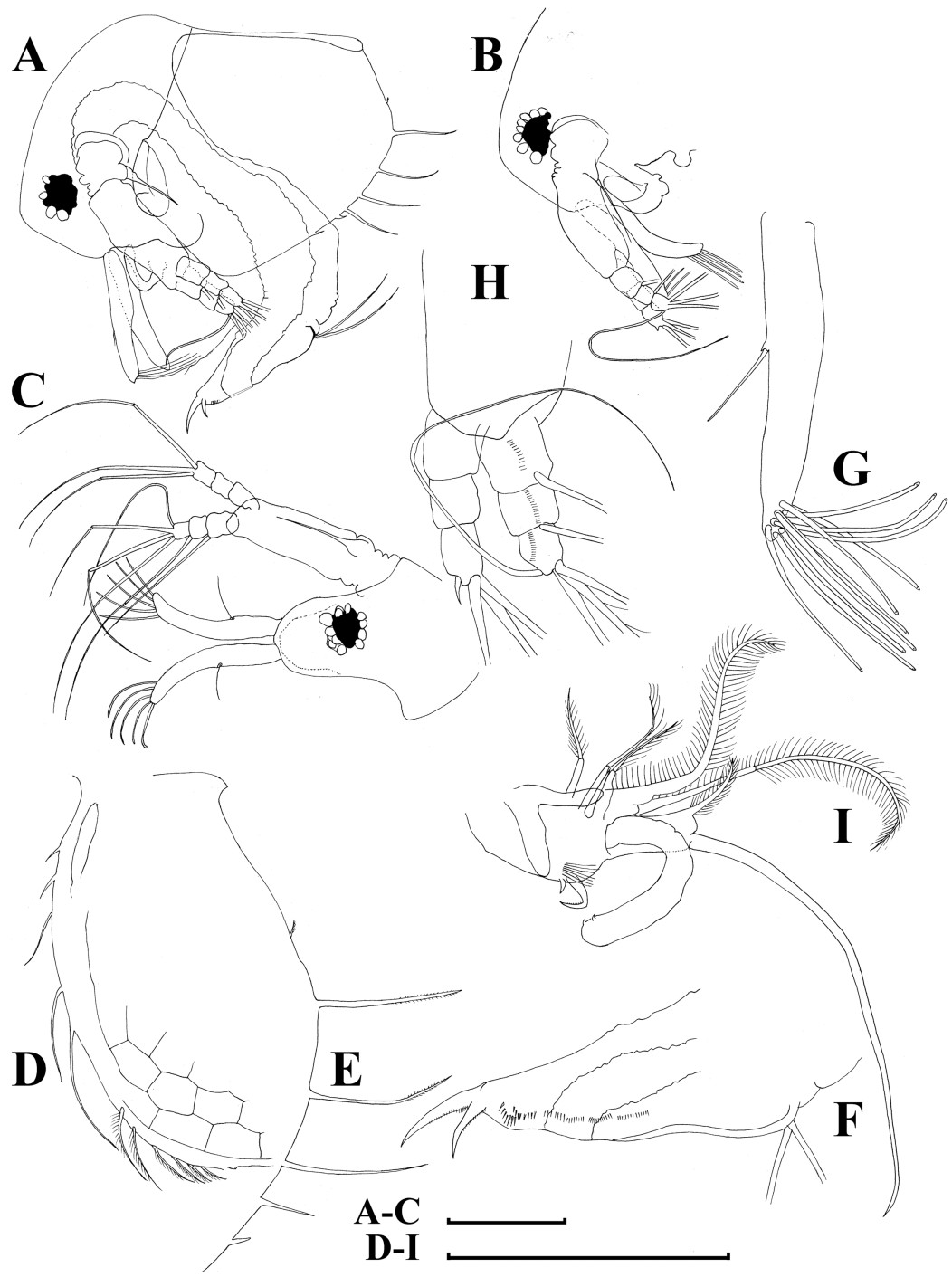

**Figure 14** *Bosminopsis zernowi Linko, 1901*, **adult male from Lake Livadijskoe, Primorski Territory, Far East of Russia.** (A), Lateral view. (B), Head, lateral view. (C), Its anterior view. (D), Valve antero-ventral portion. (E), Posterior portion of valve. (F), Postabdomen. (G), Antenna I. (H), Antenna II. (I), Limb I. Scale bars = 0.1 mm.

*Bosminopsis ishikawai Klocke, 1903. Klocke, 1903*, p. 130–134, figs 5–8, Pl. 4: figs 2, 6; *Burckhardt, 1924*, p. 222.
*Bosminopsis klockei Burckhardt, 1909. Burckhardt, 1909*, p. 251; *Burckhardt, 1924*, p. 222.

*Bosminopsis pernodi Burckhardt, 1909*. *Burckhardt, 1909*, p. 251; *Burckhardt, 1924*, p. 222.
*Bosminopsis deitersi pernodi* in *Manujlova, 1964*, p. 265.
*Bosminopsis schroeteri Burckhardt, 1909*. *Burckhardt, 1909*, p. 251; *Burckhardt, 1924*,
p. 229, Fig. 1, 4, 6–7.

**Type locality.** "Flusse Wjatka gefunden" = the Vyatka River (affluent of the Kama River which is a large affluent of Volga) near Malmyzh (*Zernov, 1901*), Kirov Area, European Russia.

**Type material.** Lost.

**Material studied here.** See Table S1.

**Short diagnosis.** Body of large adult parhenogenetic female (Figs. 10A, 11A–11B) subovoid, in younger adults more elongated, with a short postero-dorsal spine, but a caudal needle absent. Reticulation ill-expressed on valves and head (Figs. 11C–11D). Valve (Figs. 11E–11H) with a series of mucro-like spines at postero-ventral valve portion, or they are completely reduced. Postabdomen without inflated basis of postabdominal setae (Figs. 12A–12C). Antenna I and II (Figs. 11I–11J) as in previous species. Limb I with epipopide having two finger-like projections, limbs in general as in previous species (Figs. 12D–12M), but seta 1 en exopodite III relatively shorter, seta 7 there relatively longer (Fig. 12F), seta 2 on inner-distal limb portion longer than seta 1, longest setae in distal armature of gnathobase III strongly longer than other setae (Fig. 12G); on exopodite V, seta 2 and 3 short (Fig. 12M). Juvenile female (Fig. 10F–G) with a series of long, thin mucro-like spines. Free and fused parts of antennae I, mucro, rostrum, ventral valve edge, base of caudal spine covered with small spinules. Ephippial female (Figs. 10B–10D, 11K–11L) with egg chamber sculpture represented by large polygons, but this sculpture is less represented as compare too previous species. A strong medial keel on dorsum, strong paired lateral keels well distinguishable from the dorsal view. Juvenile male (Fig. 10H, Fig. 13) as in previous species. Adult male with a dorsal contour of head humped (Fig. 14A); head large (Figs. 14B–14C), with a smooth rostrum and expressed ocular dome, a series of mucro-like spines always present postero-ventrally. Valve as in female (Figs. 14D–14E). Basal spine on postabdominal claw shorter than the claw itself (Fig. 14F). Antenna I free, its distal portion only slightly bent (Fig. 14C, 14G). A long additional male seta on endopod not curved at tip (Fig. 14H). Limb I with a copulatory hook (Fig. 14I) relatively more massive that in previous species.

**Size.** Females 0.25–0.47 mm, adult males 0.26–0.30 mm.

**Differential diagnosis.** It differs from *B. deitersi* in (1) several mucro-like spines at postero-ventral valve angle in both females and males; (2) different proportions of setae in exopodite, inner limb portion and distal armature of gnathobase of limb III and on exopodite V; (3) male basal spine on postabdominal claw approximately as long as claw itself; (4) male antenna I almost not bent distally; (5) additional seta on apical segment of male antenna II without curved tip.

**Distribution.** In Europe, *B. zernowi* is recorded from the Neman basin in Poland (*Wolski, 1932*), Dniepr River basin (including Dniepr itself, Desna and Pripyat, in Ukraine and Belarus, and, most probably, the Russian portion of the basin) (*Werestchagin, 1912*; *Charleman, 1915*, *1922*; *Vezhnovets, 2005*). *Bledzki & Rybak (2016)* included the Danube basin as the part of its range, but no records from this river are known to us. Negrea (*Negrea, 1983*) wrote that the species "could be present in Romania", but to date it was not recorded from this country. In European Russia, *B. zernowi* was recorded from many rivers of the Volga basin, including the Volga itself, Kostroma, Wyatka, Kama, Kerzhenets, Sura, Kubra, Oka, Nara, and Moskva rivers (*Meissner, 1902*; *Skorikov, Bolokhontsev & Meissner, 1903*; *Zykoff, 1906*; *Greze, 1921*; *Muraveisky, 1924*; *Behning, 1928*; *Greze, 1929*; *Rylov, 1940*; *Tarbeev et al., 2011*). The species was found in the basins of all the great rivers of Western and Eastern Siberia: the Ob' River basin (*Leschinskaya, 1962*) including its Arctic portion (*Werestchagin, 1913*), the Tom' River (*Petlina et al., 2000*) and the Chulym River (*Kukharskaya & Dolgin, 2009*); Subarctic portion of the Yenisey River (*Pirozhnikov, 1937*); Lena River (*Abramova & Zhulai, 2016*), Amur River basin (*Afonina, 2013*). The opinion of Manujlova (*Manujlova*) that "in the USSR it was not found east to Ob' River" was based on inadequate knowledge of previous literature, i.e. (*Pirozhnikov, 1937*). It is widely distributed in Korea (*Cho & Mizuno, 1977*) (also see descriptions above) and Japan (*Ueno, 1937b*) (and our data), and present in South China and Vietnam (our data). Most probably, it is present on the Pacific coast of Asia from the Amur to the Mekong basins.

But previous records from China (*Ueno, 1932*; *Ueno, 1944*; *Mashiko, 1953*; Du *Nanshan, 1973*; *Chiang & Du, 1979*; *Xiang et al., 2015*) need to be checked, as they could belong both to *B. zernowi* (Amur basin) (i.e. (*Ueno, 1937a*; *Ueno, 1940*)) and the SE Asia taxon (at least some populations in southernmost China). "*B. schröteri*" described from "Sutschaufluss bei Schanghai" (*Burckhardt, 1909*), is a junior synonym of *B. zernowi* (as it has several mucro-like spines at postero-ventral angle).

Most probably, the tropical countries of Asia are fully populated by other taxa, as in all illustrations the females have a single strong mucro (*Rane, 1984*; *Idris, 1983*; *Michael & Sharma, 1988*; *Pascual et al., 2014*). Also, a single mucro is illustrated in the figures of *Bosminopsis* from North America (*Birge, 1918*), Africa (*Korinek, 1984*) and Australia (*Smirnov & Timms, 1983*) (Fig. 15).

## DISCUSSION

### Old Mesozoic group

Our results are consistent with the hypothesis that *B. deitersi* is, in fact, a species group. We find no evidence for the nominate species in the Palearctic. However, we did find evidence for a genetically divergent and morphologically differentiated Old World lineage. Notably, the strong genetic divergences that we observed and our ancient age estimates were unaccompanied by strong morphological divergences. With our integrated approach, we hoped to mitigate some of the limitations of molecular datasets. As coalescent analyses can oversplit taxa, multigene data benefit from morphological and ecological information. Single gene datasets may disagree with one another and with the

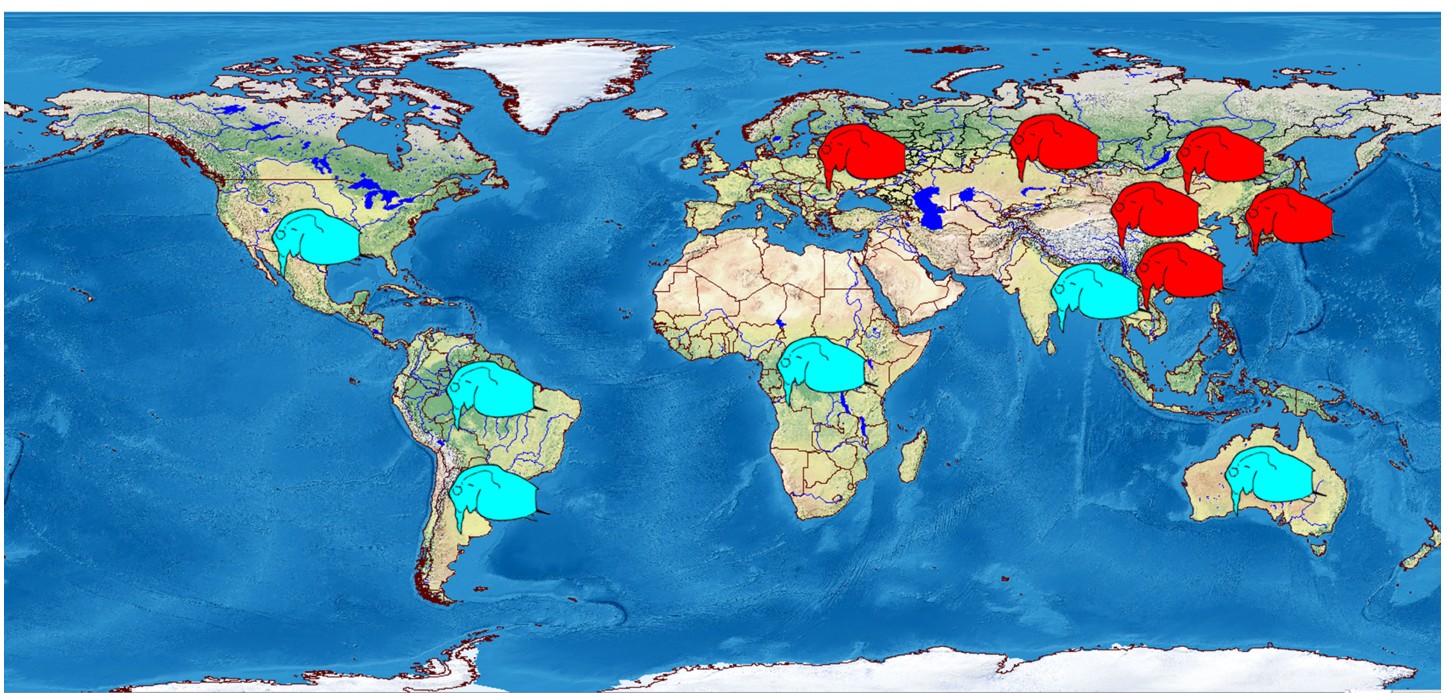

**Figure 15 Schematic representation of distribution of two major morphotypes of the juvenile parthenogenetic female: with several mucro-like spines (red) and a single mucro (blue).** Visualisation was made in free software DIVA-GIS7.5.0 https://www.diva-gis.org using free spatial GIS data from http://www.naturalearthdata.com as the layers. Symbols are inserted manually.

species tree for manifold reasons (*Fisher-Reid & Wiens, 2011*; *Hailer et al., 2012*). In the present analysis, the topological disagreements (e.g. subclades) were weakly supported, indicating that random error may play a role.

A mature biogeography is only possible with an understanding of timescale (*Rosen, 1978*). The antiquity of cladocerans of different ranks, from genera to orders, has been confirmed by the fossil record, i.e., from the Mesozoic (*Smirnov, 1992*; *Kotov & Korovchinsky, 2006*; *Kotov & Taylor, 2011*; *Liao et al., 2020*). Unfortunately, the Palaeozoic records (*Anderson, Crighton & Hass, 2004*; *Womack et al., 2012*) are dubious: the described animals could belong to the Cladocera, but also could be members of other crustacean groups (*Van Damme & Kotov, 2016*). *Kotov & Taylor (2011)* demonstrated that extant genera of the Daphniidae and even the subgenera of the genus *Daphnia* existed at the Jurassic/Cretaceous boundary, ca. 145 MYA. More fossil calibrations are possible for the group.

Efforts to use fossil calibrations with molecular data have been limited for Cladocera (*Sacherova & Hebert, 2003*; *Schwentner et al., 2013*; *Cornetti et al., 2019*). Perhaps the only known calibration point for relaxed molecular clocks is the *Daphnia/Ctenodaphnia/Simocephalus* split at 145 MYA (*Kotov & Taylor, 2011*). Non-calibrated molecular clocks also suggest earlier differentiation of the cladocerans, i.e., differentiation of the subfamilies within Chydoridae in the mid-Palaeozoic (*Sacherova & Hebert, 2003*). A fast molecular clock estimate gave a divergence time for *Daphnia* at more than 66 MYA (*Cornetti et al.,*
*2019*). This value is probably too young given the calibration point of 145 MYA. A more realistic estimate should exceed the minimum fossil calibration (*Kotov, 2013*).

The Family Bosminidae contains only the genus *Bosmina* and the genus *Bosminopsis*. Our very rough estimation (see Fig. 4A) suggests that the Bosminidae could be even older than the Daphniidae. Such a conclusion agrees with the hypothesis that bosminids are a sister group to Chydoridae (*Kotov, 2013*). Chydorids are probably of Palaeozoic origin (*Sacherova & Hebert, 2003*). No Mesozoic bosminids are known to date. Bosmina was one of the first genera to be studied with paleolimnology. Unlike *Bosmina*, subfossil remains for *Bosminopsis* are unknown from the Holocene and Pleistocene bottom sediments (*Austin, 1942*; *Hofmann, 1984*). It seems unlikely that a detailed fossil record will be found for *Bosminopsis*. So molecular clocks are the only method to estimate the time of its differentiation. We estimate that the differentiation of the main *Bosminopsis* lineages took place in the Cretaceous, and coincided with the disruption of Pangaea, or later disruption of Laurasia. Mesozoic lineages survived in SE Asia and elsewhere in Eurasia (the exact location is unclear) after a mass extinction during the mid-Caenozoic (*Korovchinsky, 2006*; *Van Damme & Kotov, 2016*). Most probably, the Pacific Coast Region ("Far East") was the center of *B. zernowi* diversificaton, as this region is the richest in mitochondrial haplotypes (Fig. 3), and is often a center of diversity for cladoceran taxa (*Kotov et al., 2021*).

While there is strong genetic divergence between New World and Old World lineages, a more detailed assessment of the divergence time awaits further geographic and genomic sampling for the New World. Within *B. zernowi*, our results suggest a mitochondrial differentiation in the mid-late Caenozoic (or even Quaternary), but the divergence of its mitochondrial haplotypes was weak.

*Korovchinsky (2006)* postulated that extant cladocerans are relicts of a mass extinction in the mid-late Caenozoic. For Cladocera, Pleistocene mass extinction in the Holarctic due to glaciation and aridization (*Hewitt, 2000*) also has phylogeographic support (*Taylor, Finston & Hebert, 1998*; *Cox & Hebert, 2001*). But phylogeographic publications referring to previous epochs with non-Holarctic samples are rare (*Xu et al., 2009*; *Kotov et al., 2021*). For *Bosminopsis*, our results suggest a Mesozoic differentiation of the lineages and then survival of only two main lineages in the mid-Caenozoic. We failed to detect divergences consistent with the Quaternary. Our results are consistent with contintental endemism and longterm morphological stasis.

Morphological divergence in *Bosminopsis* appears to be weak since the Mesozoic. This divergence involves fine-scaled characters such as the mucro-like spine number, male basal spine, and antenna I appearance and armature. Such subtle differences among species are known in other cladoceran groups (*Kotov et al., 2021*) but can only rarely be associated with a timescale.

There are no known fossil records for the globally distributed *B. deitersi* group. However, *Bosminopsis* may be a "living fossil" sensu Darwin (*Darwin, 1859*). The *B. deitersi* group has survived with very little morphological change since the Mesozoic despite profound abiotic and biotic changes to the continental water bodies over this

timescale. Our results indicate that the occupation of differing climates has also left a weak morphological signature. While the concept of "living fossil" is somewhat ambiguous (*Casane & Laurenti, 2013*) there are several groups that appear to have undergone morphological stasis since the Mesozoic. Our evidence is consistent with *Frey (1962)* who expected stasis to account for continental endemism in Cladocerans.

### Preliminary comments on further taxonomic revision

Further studies are needed to demonstrate that the North American populations form a separate species from South American specimens. If so, then the taxonomic name for North American specimens would be *Bosminopsis birgei Burckhardt, 1924*. Records of *Bosminopsis* are infrequent in North America and are mainly from the southeastern USA (*Pennak, 1953*; *Beaver et al., 2018*). Recent biotic exchange between North America and South America has occurred for several cladoceran genera (*Mergeay et al., 2008*). In such cases, there is very little genetic differentiation for mitochondrial markers among continents. The status of populations from East Asia (*Idris, 1983*; *Michael & Sharma, 1988*) must also be addressed, including that of *B. devendrari* Rane (a possible proper name for the SE taxon recorded above). To date, we have no information on *Bosminopsis* cf. *deitersi* from Africa, i.e., described by *Daday (1908)* from Lake Nyassa as *Bosminella Anisitsi* var. *africana* with a single mucro. This taxon is found in different African countries (*Brehm, 1913*; *Rahm, 1956*; *Korinek, 1984*). The status of Australian populations (*Smirnov & Timms, 1983*), also having single mucro, remains unclear.

Subtle geographic variation in the morphology of *Bosminopsis* has been known since the early 1900s. *Meissner (1903)* and *Klocke (1903),* for example, pointed out the numerous stout spines at the postero-ventral corner of the valves from Russian and Japanese *Bosminopsis*–the most prominent difference between *B. deitersi* and *B. zernowi*. Still, subsequent authors failed to recognize this variation as taxonomically valuable. *Klocke (1903)* concluded that there are two species in Japan, *B. ishikawai* and *B. deitersi*. He concluded that the former has stronger denticles on antenna I, better-developed reticulation, a postero-dorsal projection located more dorsally and longer spines at postero-dorsal angle "making it similar to *Ilyocryptus*". In reality, all listed differences are characteristic of juvenile females. Therefore, *Klocke (1903)* erroneously regarded the populations with large adults and without large adults those as two separate species. The same mistake was made by *Rey & Vasquez (1986),* who described *B. macaguensis* referring to differences of juvenile males of *B. deitersi* in Venezuelan populations (*Kotov, 1997a*).

## CONCLUSIONS

Here we revised only populations from Eurasia. Other taxa discussed above in the genetic section must be reconsidered in the future, and biological differences must be studied in detail. Each of these putative species has a single mucro at a postero-dorsal angle and minimal differences between parthenogenetic females. We expect that comparing males will be the most fruitful for assessing morphological diagnoses, as male morphology tends

to differ among species more than female morphology in cladocerans (*Popova et al., 2016*; *Sinev, Karabanov & Kotov, 2020*).

## ACKNOWLEDGEMENTS

Many thanks to E.S. Chertoprud, D.E. Gavrilko, L. Hovind, S. Ishida, H.G. Jeong, N.M. Korovchinsky, W.H. Piel, D.E. Shcherbakov, A.Y. Sinev for zooplankton samples, K.S. Chae, H.J. Dumont, B.P. Han, S.J. Ji, H.S. Kim for assistance during sampling. SEM works are carried out at the Joint Usage Center "Instrumental Methods in Ecology" at A.N. Severtsov Institute of Ecology and Evolution of Russian Academy of Sciences.

### Funding

This study was supported exclusively by the Russian Science Foundation (grant no. 18-14-00325 for Petr G. Garibian, Dmitry P. Karabanov, Anna N. Neretina, and Alexey A. Kotov). Sampling in South Korea preceding the project was supported by a grant from the National Institute of Biological Resources (NIBR), funded by the Ministry of Environment (MOE) of the Republic of Korea for Alexey A. Kotov. Derek J. Taylor has no specific support. The funders had no role in study design, data collection and analysis, decision to publish, or preparation of the manuscript.

### Grant Disclosures

The following grant information was disclosed by the authors:
Russian Science Foundation: 18-14-00325.
National Institute of Biological Resources (NIBR).
Ministry of Environment (MOE) of the Republic of Korea.

### Competing Interests

The authors declare that they have no competing interests.

### Author Contributions

- Petr G. Garibian conceived and designed the experiments, performed the experiments, prepared figures and/or tables, authored or reviewed drafts of the paper, and approved the final draft.
- Dmitry P. Karabanov conceived and designed the experiments, performed the experiments, analyzed the data, prepared figures and/or tables, authored or reviewed drafts of the paper, and approved the final draft.
- Anna N. Neretina performed the experiments, prepared figures and/or tables, sEM studies, and approved the final draft.
- Derek J. Taylor analyzed the data, authored or reviewed drafts of the paper, and approved the final draft.
- Alexey A. Kotov conceived and designed the experiments, analyzed the data, prepared figures and/or tables, authored or reviewed drafts of the paper, and approved the final draft.

## Field Study Permissions

The following information was supplied relating to field study approvals (i.e., approving body and any reference numbers):

Field collection in public property in Russia does not require permissions. Samples in South Korea were collected in the frame of cooperation between A.A. Kotov and the National Institute of Biological Resources of Korea and does not require special permission. The sample from Arkansas, USA was obtained from collections resulting from a previous NSF grant. The samples from Japan, China, and Thailand were provided by our colleagues having permissions to collect them due to their scientific activity in the governmental institutes in the corresponding countries. Formol samples from Brazil were kept in the Collection of Zoological Museum of Moscow State University for a long time, they were collected before the time when Brazil introduced very strict regulations for sampling. The species were not assessed as endangered at the time of collection and are currently not subject to specific regulations, however all efforts were taken to ensure that the collection and preservation of animals was performed with due consideration of their welfare. The number of individuals taken did not represent a significant proportion of the population present at each site.

## DNA Deposition

The following information was supplied regarding the deposition of DNA sequences:

All data generated or analysed during this study are included in Open Science Framework project (https://osf.io/4fjnm/) and this published article. All sequences are deposited at the NCBI GenBank accs. no. MT757174–MT757274, MT757314–MT757388, MT757459–MT757473. Specimen series from which DNA was extracted are deposited at Zoological Museum of Moscow State University.

## Data Availability

Samples are deposited at the Collection of Zoological Museum of Moscow University, Moscow, Russia:

MGU Ml 211_*Bosminopsis zernowi*_10_Russia (Asian)_Primorski Territory_Lake Livadijskoe, Nakhodka Area

MGU Ml 212_*Bosminopsis zernowi*_11_Japan__Sai-no-Kami Ike, Ise County, Mie

MGU Ml 213_Bosminopsis zernowi_4_Japan_Toyama_Oh-Zuka Ike

MGU Ml 214_*Bosminopsis zernowi*_2_Japan_Toyama_Oh-Zuka Ike

MGU Ml 215_*Bosminopsis zernowi*_9_Russia (Asian)_Sakhalin Area_Russkoe Lake, Puzina Peninsula

MGU Ml 216_*Bosminopsis deitersi*_1_China__Jiu Lake

MGU Ml 217_*Bosminopsis zernowi*_18_Japan_Shiga_Sho-Jyo Ko

MGU Ml 218_*Bosminopsis zernowi*_18_Japan_Mie_Sai-no-Kami Ike

MGU Ml 219_*Bosminopsis zernowi*_10_Russia (Asian)_Primorski Territory_Ilyinskoe Lake 1 (near the road),region of Khanka Lake

MGU Ml 220_*Bosminopsis zernowi*_11_Russia (Asian)_Primorski Territory_Ilyinskoe Lake 2, region of Lake Khanka

MGU Ml 221_*Bosminopsis deitersi*_12_Thailand_Kalasin Province_Bung Pueng

MGU Ml 222_*Bosminopsis zernowi*_10_South Korea_Gyeongsangnam-do_Hap Cheon 1 (Reservoir)

MGU Ml 223_*Bosminopsis zernowi*_2_Russia (Asian)_Yakutia Autonomous Republic_Kuria Strelka, town of Yakutsk

MGU Ml 224_*Bosminopsis zernowi*_5_Russia (Asian)_Yakutia Autonomous Republic_A River Lena affluent near Kurja Strelka, Yakutsk

MGU Ml 225_*Bosminopsis zernowi*_10_Russia (Asian)_Primorski Territory_A quarry near Borisovsky Most

MGU Ml 226_*Bosminopsis zernowi*_1_Russia (Asian)_Primorski Territory_Roadside pond

MGU Ml 227_*Bosminopsis* cf. *deitersi*_10_Vietnam__Huang River, Hue

MGU Ml 228_*Bosminopsis zernowi*_1_South Korea_Jeollanam-do_Jaeun reservoir

MGU Ml 229_*Bosminopsis zernowi*_10_South Korea_Gyeongsangbuk-do_Hagok reservoir (a large, shallow lake, no littoral zone)

MGU Ml 230_*Bosminopsis zernowi*_10_South Korea_Gyeongsangbuk-do_Hagok reservoir (a large, shallow lake, no littoral zone)

MGU Ml 231_*Bosminopsis zernowi*_10_South Korea_Jeollanam-do_Useul reservoir

MGU Ml 232_*Bosminopsis zernowi*_10_South Korea_Jeollanam-do_A pond near Yanghwa Town Hall

MGU Ml 233_*Bosminopsis deitersi*_2_Brazil_Amazonas_Lago do Castanho

MGU Ml 234_*Bosminopsis zernowi*_14_Ukraine_Poltava Area_a tributary of Dnepr River near Kremenchug

MGU Ml 235_*Bosminopsis deitersi*_16_Brazil_Pará_Rio Tapajóz, b. Fordlandia, Zool.-Oberfl.

Data available at Open Science Framework project: Karabanov, Dmitry P, and Alexey A Kotov. 2020. "Bosminopsis Deitersi Complex." OSF. Dataset. https://osf.io/4fjnm/.

## Supplemental Information

Supplemental information for this article can be found online at http://dx.doi.org/10.7717/peerj.11310#supplemental-information.

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
