# Peer review of "Bosminopsis deitersi (Crustacea: Cladocera) as an ancient species group: a revision"

_PeerJ, doi:10.7717/peerj.11310_

## Round 0.1 · original submission · Minor Revisions

I have heard back from two reviewers, both of whom have offered constructive comments that should help you improve this work. I look forward to seeing a new version in the near future.

Reviewer 1 ·

Basic reporting

Adequate

Experimental design

Adequate

Validity of the findings

Adequate

Additional comments

The manuscript titled “Bosminopsis deitersi (Crustacea: Cladocera) as an ancient species group: a revision” bring important information about biogeography, taxonomy and cryptic diversity of the Bosminopsis genus. Manuscript contents indicate the high level of authors knowledge about the theme. In my opinion, the manuscript should be accepted with minor revision.

#Introduction
The objectives might be improved if the authors indicate (1) the redescription of B. deitersi and comparison with B. zernowi, (2) also the analysis of synonymous.

#Materials and Methods
It is not clear if specimens fixed in formalin were used in molecular analysis. If it was, which treatment was performed on the samples before DNA extraction?

#Results (Description)
General – The drawings of appendages need of more elaboration to elucidate homology issues because numeration of exopodite setae is lacking.

Limb II – seems that proportion of setae on anterior and posterior portion in booth species are different. This need be described and used on the diagnoses and diagnoses differential.

Limb III - seems that proportion of setae on exopodite in booth species are different. The same to setae 1-2 on the distal endite. This need be described and used on the diagnoses and diagnoses differential.

Limb V - seems that proportion of setae on exopodite in booth species are different. The same to setae on the distal portion. This need be described and used on the diagnoses and diagnoses differential.

·

Basic reporting

Authors did not provide measurements of different sexes and ages in studied populations, giving only size range for whole population ("Size. 0.17–0.41 mm"). But males of cladocera are always smaller than females, and size range of adult male is an important taxonomical character! I suggest authors should produce separately measurements of females, including minimum size of ovigerous female, and measurements of juvenile and adult males.

Authors did not refer to monograph of Idris (1983) on cladocera of Malaysia, which contains quite detailed description of local Bosminopsis population, with well-developed mucro.

Experimental design

No comments.

Validity of the findings

No comments.

Additional comments

Through the document, works of Van Damme are referenced as “van Damme”, this is not correct, Van Damme is a family name of two words.

Lines 429-430 "No postero-dorsal spine (caudal needle) on carapace and inflated basis of postabdominal setae."
The statement is unclear and should be reworked, probably split into two sentences.

---

## Round 0.2 · Minor Revisions

Thank you for your revision; it is very well done and I am almost ready to accept this work. I have added two small edits to your work, and request that you include the details of molecular analyses in the Materials and Methods instead of simply referring to Rostov et al. 2021. You can still reference this paper too, but as we are not constricted by page restrictions here, I strongly prefer keeping these details. Please see the attached PDF for details.

I imagine edits should take no more than an hour at most, and look forward to seeing the final version of your work.

Reviewer 1 ·

Basic reporting

Adequate

Experimental design

Adequate

Validity of the findings

Adequate

Additional comments

Dear editor,

the authors modified the manuscript according to suggestions of reviwers. Thus, the manuscript should be accepted for publication.

My congratulations to authors!

---

## Round 0.3 · accepted · Accept

I am happy with the edits, and very pleased to move this into production. Please note there is one reference on lines 245-246 (Место для ввода текста.) that needs to be edited.